**EMBO** *reports*

# Structural basis for Mis18 complex assembly and its implications for centromere maintenance

Reshma Thamkachy[1,8], Bethan Medina-Pritchard[1,8], Sang Ho Park[2,8], Carla G Chiodi[1], Juan Zou[1], Maria de la Torre-Barranco[1], Kazuma Shimanaka[2], Maria Alba Abad[1], Cristina Gallego Páramo[1], Regina Feederle[3], Emilija Ruksenaite[4], Patrick Heun[1], Owen R Davies[1], Juri Rappsilber[1,5], Dina Schneidman-Duhovny[6], Uhn-Soo Cho[2] & A Arockia Jeyaprakash[1,7]✉

## Abstract

**The centromere, defined by the enrichment of CENP-A (a Histone H3 variant) containing nucleosomes, is a specialised chromosomal locus that acts as a microtubule attachment site. To preserve centromere identity, CENP-A levels must be maintained through active CENP-A loading during the cell cycle. A central player mediating this process is the Mis18 complex (Mis18α, Mis18β and Mis18BP1), which recruits the CENP-A-specific chaperone HJURP to centromeres for CENP-A deposition. Here, using a multi-pronged approach, we characterise the structure of the Mis18 complex and show that multiple hetero- and homo-oligomeric interfaces facilitate the hetero-octameric Mis18 complex assembly composed of 4 Mis18α, 2 Mis18β and 2 Mis18BP1. Evaluation of structure-guided/separation-of-function mutants reveals structural determinants essential for cell cycle controlled Mis18 complex assembly and centromere maintenance. Our results provide new mechanistic insights on centromere maintenance, high-lighting that while Mis18α can associate with centromeres and deposit CENP-A independently of Mis18β, the latter is indispensable for the optimal level of CENP-A loading required for preserving the centromere identity.**

**Keywords** Centromere; CENP-A; Mis18 Complex; Centromere Inheritance; Integrative Structural Analysis
**Subject Categories** Cell Cycle; Chromatin, Transcription & Genomics; Structural Biology

## Introduction

Faithful chromosome segregation during cell division requires the bi-orientation of chromosomes on the mitotic spindle through the physical attachment of kinetochores to microtubules.

Kinetochores are large multiprotein scaffolds that assemble on a special region of chromosomes known as the centromere (Catania and Allshire, 2014; Cheeseman, 2014; Fukagawa and Earnshaw, 2014; Musacchio and Desai, 2017). Whilst centromeres in some organisms, such as budding yeast, are defined by a specific DNA sequence, in most eukaryotes, centromeres are distinguished by an increased concentration of nucleosomes containing a histone H3 variant called CENP-A (Black et al, 2010; Fukagawa and Earnshaw, 2014; McKinley and Cheeseman, 2016; Stellfox et al, 2013). CENP-A containing nucleosomes recruit CENP-C and CENP-N, two proteins that are part of the constitutive centromere-associated network (CCAN) and that recruit the rest of the kinetochore components at the centromeric region of the chromosome (Carroll et al, 2010; Kato et al, 2013; Weir et al, 2016).

Whilst canonical histone loading is coupled with DNA replication, CENP-A loading is not (Dunleavy et al, 2011). This results in a situation where, after S-phase, the level of CENP-A nucleosomes at the centromere is halved due to the distribution of existing CENP-A to the duplicated DNA (Dunleavy et al, 2009; Jansen et al, 2007). To maintain centromere identity, centromeric CENP-A levels must be restored. This is achieved through active CENP-A loading at centromeres (during G1 in humans) via a pathway that requires the Mis18 complex (consisting of Mis18α, Mis18β and Mis18BP1) and the CENP-A chaperone, HJURP (Barnhart et al, 2011; Dunleavy et al, 2009; Foltz et al, 2009; Fujita et al, 2007; Jansen et al, 2007) (Fig. 1A). The Mis18 complex can recognise and localise to the centromere, possibly through its proposed binding to CENP-C and/or other mechanisms which have not yet been identified (Dambacher et al, 2012; Moree et al, 2011; Stellfox et al, 2016). Once at the centromere, the Mis18 complex has been implicated in facilitating the deposition of CENP-A in several ways. There is evidence that the Mis18 complex affects DNA methylation and histone acetylation, which may facilitate CENP-A loading (Hayashi et al, 2004; Kim et al, 2012). But one of the most important and well-established roles of the Mis18 complex is the recruitment of HJURP, which binds a single

[1]Wellcome Centre for Cell Biology, University of Edinburgh, Edinburgh EH9 3BF, UK. [2]Department of Biological Chemistry, University of Michigan, Ann Arbor, MI 48109, USA. [3]Monoclonal Antibody Core Facility, Helmholtz Zentrum München, German Research Center for Environmental Health (GmbH), 85764 Neuherberg, Germany. [4]Institute Novo Nordisk Foundation Centre for Protein Research, Copenhagen, Denmark. [5]Institute of Biotechnology, Technische Universität Berlin, 13355 Berlin, Germany. [6]School of Computer Science and Engineering, The Hebrew University of Jerusalem, Jerusalem, Israel. [7]Gene Center, Department of Biochemistry, Ludwig Maximilians Universität, Munich, Germany. [8]These authors contributed equally: Reshma Thamkachy, Bethan Medina-Pritchard, Sang Ho Park. ✉E-mail: jeyaprakash.arulanandam@ed.ac.uk

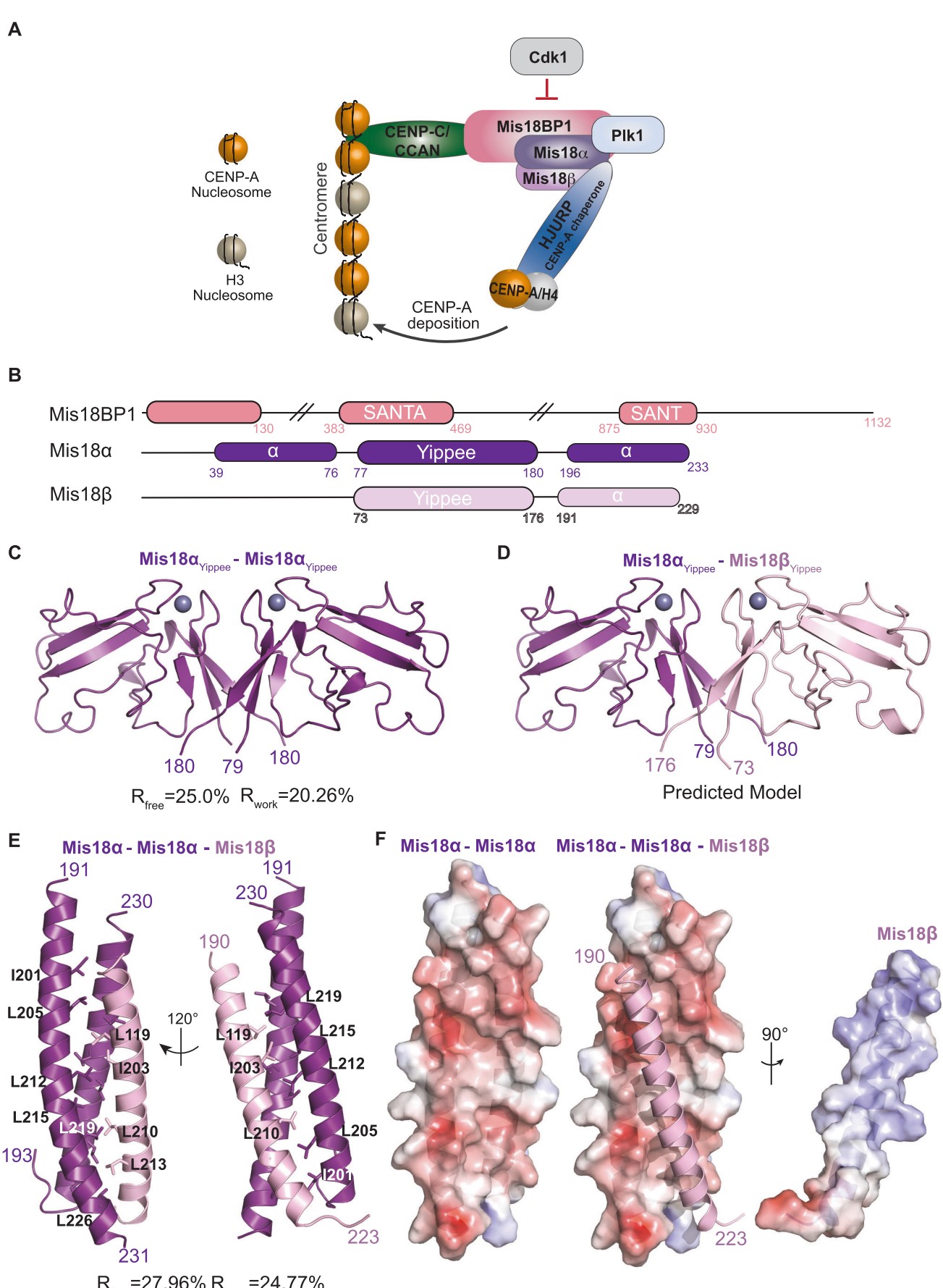

**Figure 1.   Mis18α/β contains two independent structural domains that can oligomerise.**

(A) Diagram of proteins involved in CENP-A deposition at the centromere. The Mis18 complex (Mis18BP1 (salmon), Mis18α (purple) and Mis18β (light pink)) forms once Cdk1 activity is reduced. It interacts with CCAN/CENP-C (green) to localise to the centromere, where Plk1 regulation helps promote the recruitment of HJURP (Blue), a CENP-A chaperone. (B) Schematic representation of structural features of Mis18BP1 (salmon), Mis18α (purple) and Mis18β (light pink). Filled boxes represent folded domains. SANTA and SANT domain boundaries as defined in UniProt (Q6P0N0). (C) Cartoon representation of the crystal structure of human Mis18α_Yippee homodimer (PDB ID: 7SFZ). (D) Cartoon representation of the human Mis18α_Yippee/Mis18β_Yippee heterodimer modelled by homology to the structure in Fig. 1C. Mis18α is shown in purple and Mis18β in light pink (modelled using Phyre2, www.sbg.bio.ic.ac.uk/phyre2/ (Kelley et al, 2015)). (E) Cartoon representation of the crystal structure of Mis18α_C-term/Mis18β_C-term (PDB ID: 7SFY). Mis18α is shown in purple and Mis18β in light pink. (F) Mis18α_C-term domains are shown in surface representation and coloured based on electrostatic surface potential calculated using APBS (Baker et al, 2001). Mis18β_C-term shown as cartoon.

CENP-A/H4 dimer and brings it to the centromere (Barnhart et al, 2011; Dunleavy et al, 2009; Hu et al, 2011). This then triggers a poorly understood process in which the H3 nucleosomes are removed and replaced with CENP-A nucleosomes. Finally, the new CENP-A nucleosomes are stably integrated into the genome, which requires several remodelling factors such as MgcRacGAP, RSF, Ect2 and Cdc42 (Lagana et al, 2010; Perpelescu et al, 2009).

The timing of CENP-A deposition is tightly regulated, both negatively and positively, by the kinases Cdk1 and Plk1, respectively, in a cell cycle-dependent manner (McKinley and Cheeseman, 2014; Muller et al, 2014; Pan et al, 2017; Silva et al, 2012; Spiller et al, 2017; Stankovic et al, 2017). Previous studies demonstrated that Cdk1 phosphorylation of Mis18BP1 prevents the Mis18 complex assembly and localisation to centromeres until the end of mitosis (when Cdk1 levels are reduced) (Pan et al, 2017; Spiller et al, 2017). Cdk1 also phosphorylates HJURP, which negatively regulates its binding to the Mis18 complex at the centromere (Muller et al, 2014; Stankovic et al, 2017; Wang et al, 2014). In cells, Plk1 is a positive regulator, and its activity is required for G1 centromere localisation of the Mis18 complex and HJURP. Plk1 has been shown to not only phosphorylate Mis18α/β and Mis18BP1, but it has also been proposed to interact with phosphorylated Mis18 complex through its polo-box domain (PBD) (McKinley and Cheeseman, 2014).

As outlined above, a central event in the process of CENP-A deposition at centromeres is the Mis18 complex assembly. The Mis18 proteins, Mis18α and Mis18β, possess a well-conserved globular domain called the Yippee domain (also known as the MeDiY domain; spanning residues 77–180 in Mis18α and 73–176 in Mis18β) and C-terminal α-helices (residues 196–233 in Mis18α and 191–229 in Mis18β). We and others previously showed that the Yippee domains of Mis18 proteins can form a heterodimer, while the C-terminal helices form a heterotrimer with two Mis18α and one Mis18β. However, the full-length proteins form a hetero-hexameric assembly with 4 Mis18α and 2 Mis18β. This led to a proposed model, where the Mis18α and Mis18β mainly interact via the C-terminal helices to form a heterotrimer, and two such heterotrimers interact via the Yippee heterodimerisation (Mis18α/Mis18β) or/and homodimerisation (Mis18α/Mis18α) to form a hetero-hexameric assembly (Nardi et al, 2016; Pan et al, 2017; Pan et al, 2019; Spiller et al, 2017).

Mis18BP1, the largest subunit of the Mis18 complex (1132 aa residues), is a multi-domain protein containing SANTA (residues 383–469) and SANT (residues 875–930) domains, which are known to have roles in regulating chromatin remodelling (Aasland et al, 1996; Maddox et al, 2007; Zhang et al, 2006). In between these two domains resides the CENP-C binding domain (CBD) (Dambacher

et al, 2012; Stellfox et al, 2016). In vivo, the CBD alone is not sufficient to recruit Mis18BP1 to the centromere and requires the N-terminus of the protein for proper localisation (Stellfox et al, 2016). We and others have previously shown that the N-terminal 130 amino acids of Mis18BP1 are sufficient for interaction with Mis18α/β through their Yippee domains, and Cdk1 phosphorylation of Mis18BP1 at residues T40 and S110 inhibits its interaction with Mis18α/β to form an octamer complex consisting of 2 Mis18BP1, 4 Mis18α and 2 Mis18β (Pan et al, 2017; Spiller et al, 2017). Perturbing the Yippee domain-mediated hexameric assembly of Mis18α/β (that resulted in a Mis18α/β heterotrimer, 2 Mis18α and 1 Mis18β) abolished its ability to bind Mis18BP1 in vitro and in cells (Spiller et al, 2017), emphasising the requirement of maintaining correct stoichiometry of Mis18α/β subunits. Consistent with this, artificial dimerisation of Mis18BP1, by expressing Mis18BP1 as a GST-tagged protein, enhanced the centromere localisation of Mis18BP1 (Pan et al, 2017).

Although the importance of the Mis18 complex assembly and function is well-appreciated, a structural understanding of the intermolecular interfaces responsible for the Mis18 complex assembly and their functions is yet to be identified. Here, we have characterised the structural basis of the Mis18 complex assembly using an integrative structure modelling approach that combines X-ray crystallography, Electron Microscopy (EM), Small Angle X-ray Scattering (SAXS), Cross-Linking Mass Spectrometry (CLMS), AlphaFold and computational modelling. By evaluating the structure-guided mutations in vitro and in vivo, we provide important insights into the key structural elements responsible for Mis18 complex assembly and centromere maintenance.

## Results

### Structural basis for the assembly of Mis18α/β core modules

Mis18α and Mis18β possess two distinct but conserved structural entities, a Yippee domain and a C-terminal α-helix (Figs. 1B and EV1A,B). Mis18α possesses an additional α-helical domain upstream of the Yippee domain (residues 39–76) as compared with Mis18β. Previous studies have shown that Mis18α Yippee domain can form a homodimer or a heterodimer with Mis18β Yippee domain whereas Mis18α/β C-terminal helices form a robust 2:1 heterotrimer (Pan et al, 2017; Spiller et al, 2017; Subramanian et al, 2016). Disrupting Yippee homo- or heterodimerisation in full-length proteins, while did not abolish their ability to form a complex, did perturb the dimerisation of Mis18α/β heterotrimer (Spiller et al, 2017). Contrarily, intermolecular interactions involving the

C-terminal helices of Mis18α and Mis18β are essential for Mis18α/β complex assembly (Nardi et al, 2016). Overall, the available biochemical data suggest the presence of at least three independent structural core modules within the Mis18α/β complex: the Mis18α Yippee homodimer, the Mis18α/β Yippee heterodimer and the Mis18α/β C-terminal helical assembly. Here, we structurally characterised these modules individually and together as a holo-complex.

### Mis18α Yippee homodimer

We previously determined a crystal structure of the Yippee domain in the only homologue of Mis18 in *S. pombe* (PDB: 5HJ0), showing that it forms a homodimer (Subramanian et al, 2016). To determine the structure of human Mis18 Yippee domains, we purified and crystalised Mis18α$_{Yippee}$ (residues 77–190). The crystals diffracted X-rays to about 3 Å resolution, and the structure was determined using the molecular replacement method. The final model was refined to R and R$_{free}$ factors of 20.26% and 25.00%, respectively (Table EV1; Fig. 1C, PDB ID: 7SFZ). The overall fold of the Mis18α$_{Yippee}$ is remarkably similar to the previously determined *S. pombe* Mis18$_{Yippee}$ homodimer structure with a RMSD of 0.92 Å (Subramanian et al, 2016). In brief, the monomeric Mis18α$_{Yippee}$ is formed by two antiparallel β-sheets that are held together by a Zn$^{2+}$ ion coordinated via loops containing C-X-X-C motifs. The Mis18α$_{Yippee}$ dimerisation is mediated via a back-to-back arrangement of a 'three-stranded' β-sheet from each monomer.

### Mis18α/β Yippee heterodimer

As repeated efforts to crystallise the Mis18α/β$_{Yippee}$ heterodimer were not successful, using the Mis18α$_{Yippee}$ as a template we generated high-confidence structural models for the Mis18α/β$_{Yippee}$ heterodimer using Raptorx (http://raptorx6.uchicago.edu/) (Källberg et al, 2012) and AlphaFold2 (Jumper et al, 2021) (Figs. 1D and EV1C,D). As observed for Mis18α$_{Yippee}$ homodimer, the Mis18α/β$_{Yippee}$ heterodimerisation is also mediated via the back-to-back arrangement of the three-stranded beta sheets of Mis18α and Mis18β Yippee domains.

### Mis18α/β C-terminal helical assembly

Previous studies have shown that recombinantly purified C-terminal α-helices of Mis18α and Mis18β form a heterotrimer with two copies of Mis18α and one copy of Mis18β (Pan et al, 2017; Spiller et al, 2017). However, in the absence of high-resolution structural information, how Mis18 C-terminal helices interact to form a heterotrimer and how the structural arrangements of α-helices influence the relative orientations of the Yippee domains, and hence the overall architecture of the Mis18α/β hexamer assembly, remained unclear. We purified Mis18α spanning aa residues 191 to 233 and Mis18β spanning aa residues 188 and 229 (Figs. 1B and EV1A,B) and crystallised the reconstituted complex. The crystals diffracted X-rays to about 2.5 Å resolution. The structure was determined using single-wavelength anomalous dispersion method. After iterative cycles of refinement and model building, the final model was refined to R and R$_{free}$ factors of 24.77% and 27.96%, respectively (Table EV1, PDB ID: 7SFY). The asymmetric unit contained two copies of Mis18α/β heterotrimer. The final model included Mis18α residues 191–231 in one copy, Mis18α residues 193 to 230 in the second copy, and Mis18β residues 190–223 (Fig. 1E). The two Mis18α helices interact in an antiparallel orientation, and one helix is stabilised in a slightly curved conformation. This arrangement results in a predominantly negatively charged groove that runs diagonally on the surface formed by the Mis18α helices (Fig. 1E,F). In contrast, the pI of the Mis18β helix is 8.32. This charge complementarity appears to facilitate the interaction with Mis18α, as a positively charged surface of the Mis18β helix snugly fits in the negatively charged groove of the Mis18α/α interface. A closer look at the intermolecular interactions reveals tight hydrophobic interactions along the 'spine' of the binding groove with electrostatic interactions 'zipping-up' both sides of the Mis18β helix (Fig. 1F). The binding free energy calculated based on the buried accessible surface area suggests a nanomolar affinity interaction between the helices of Mis18α and Mis18β. It should be noted that the crystal structure presented here differs from the previously predicted models in terms of either the subunit stoichiometry (Nardi et al, 2016) or the directional arrangement of individual subunits (Mis18α and Mis18β in parallel orientation with the 2nd Mis18α in an antiparallel orientation (this work) vs all parallel (Pan et al, 2019)). Although the Pan et al, 2019 model presented the 2nd Mis18α in a parallel orientation, they did not rule out the possibility of this assembling in an antiparallel orientation within the Mis18α/β C-terminal helical assembly (Pan et al, 2019).

## Multiple surfaces of Mis18α/β Yippee heterodimers contribute to the overall oligomeric assembly of the Mis18 complex

Full-length Mis18α/β complex or the Mis18$_{core}$ complex (Mis18α–Mis18β–Mis18BP1$_{20-130}$) were not amenable for structural characterisation using X-ray crystallography possibly due to their intrinsic flexibility. Consistent with this notion, the SAXS profiles collected for the Mis18α/β ΔN (Mis18α residues 77-End and Mis18β residues 56-End), Mis18α/β and Mis18$_{core}$ complexes suggest that these complexes possess an elongated shape with flexible features (Fig. EV2; Table EV2). Hence, to understand the overall assembly of the Mis18 complex, we took an integrative structure modelling approach, combining the crystal structures of Mis18α$_{Yippee}$ dimer and Mis18α/Mis18β C-terminal hetero-trimeric helical assembly together with the homology/AlphaFold modelling of Mis18α$_{Yippee}$/Mis18β$_{Yippee}$ heterodimer, negative staining EM, SAXS and CLMS analysis of the Mis18$_{core}$ complex.

The negative staining electron micrographs of the Mis18$_{core}$ complex cross-linked using GraFix (Kastner et al, 2008) revealed a good distribution of particles (Fig. EV3A). Particle picking, followed by a few rounds of 2D classifications, revealed classes with defined structural features (Fig. 2D). Some of the 2D projections resembled the shape of a 'handset' of a telephone with bulkier 'ear' and 'mouth' pieces. Differences in the relative orientation of bulkier features of the 2D projection suggested conformational heterogeneity (Figs. 2E and EV3B). The three-dimensional volumes calculated for the particles were similar (~220 × 105 × 80 Å) and in agreement with the D$_{max}$ calculated from SAXS analysis (Fig. EV2D). Consistent with these models, when we compared the theoretical SAXS scattering curve with the experimentally measured one for Mis18α/β ΔN, we observed a good match with $x^2$ value of 1.36 (Fig. 2F).

We attempted to assemble the whole Mis18 complex using AlphaFold-multimer (AF2M), with full-length Mis18α (in purple), Mis18β (in pink) and two small region of Mis18BP1 (20–51 and

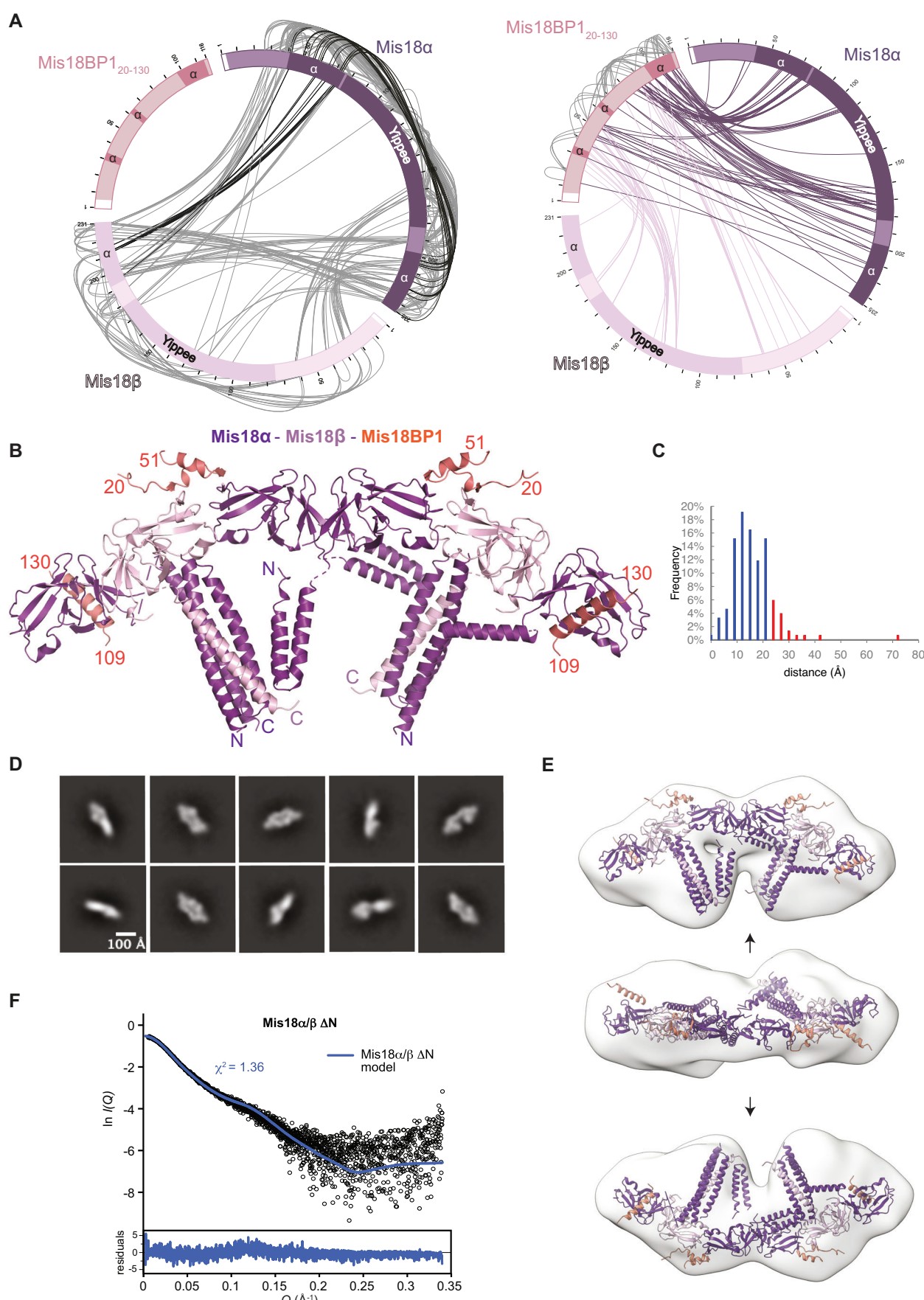

Figure 2. Mis18 complex oligomeric assembly requires multiple surfaces.

(A) Linkage map showing the sequence position and cross-linked residue pairs between the different Mis18$_{core}$ complex subunits, Mis18α, Mis18β and Mis18BP1$_{20-130}$. Left panel highlights cross-linked residues between Mis18α and Mis18β. Black lines highlight cross-links between N-terminal α-helix of Mis18α and C-terminal helical regions of proteins. Right panel highlights cross-links observed between (i) Mis18BP1$_{20-130}$ and Mis18α (purple), (ii) Mis18BP1$_{20-130}$ and Mis18β (light pink), (iii) Mis18BP1$_{20-130}$ self cross-links (light grey). White boxes represent residual residues left over from tag cleavage. Dark boxes show Yippee domains and regions of α-helices. (B) Model of the Mis18$_{core}$ complex generated using partial structures determined using X-ray crystallography and AlphaFold2 (Jumper et al, 2021) and cross-linking restrained molecular docking in EM maps. Mis18BP1 shown in salmon, Mis18α in purple and Mis18β in light pink. (C) Histograms show the percentage of satisfied or violated cross-links for structures modelled using MODELLER (Sali and Blundell, 1993). (D) Representative images of 2D classes from Mis18$_{core}$ particles picked using CryoSPARC (Punjani et al, 2017). Scale bar shows 100 Å. (E) Model (Class I) generated for Mis18$_{core}$ from negative staining EM analysis. This shows that the overall shapes of the Mis18$_{core}$ resemble a telephone handset with 'ear' and 'mouth' pieces. Arrows denote the different orientations shown. (F) Theoretical SAXS scattering curves of Mis18α/β ΔN model compared to experimental data.

109–130; in salmon) (preprint: Evans et al, 2022). The AF2M converged towards a structure with six Yippee domains stacked in a line-like arrangement in the Mis18α$_{Yippee}$-Mis18β$_{Yippee}$-Mis18α$_{Yippee}$-Mis18α$_{Yippee}$-Mis18β$_{Yippee}$-Mis18α$_{Yippee}$ order and two triple helix bundles, each formed by C-terminal α-helices of 2 copies of Mis18α and 1 copy of Mis18β. However, the modelled two helical bundles had all three helices in a parallel orientation that is not supported by our crystal structures (Fig. 1E) and cross-links (Fig. 2A). We modified the relative orientation of the helices to match the crystal structure by superposing the latter on the AF2M model (Figs. 2B,C and EV3B). Using cross-links and docking we added the N-terminal helices of the Mis18α. Cross-linking data indicates that these helices have multiple orientations with respect to the rest of the structure, contacting both Yippee domains and triple helix bundles. The linker between the Yippee domain and the C-terminal helix is the shortest in Mis18β (Fig. 1B), further supporting the arrangement of the Yippee domains within the assembly. The integrative model of the Mis18 complex fits well in the EM map. Interestingly, the serial arrangement of the Yippee domains utilises the second Yippee dimerisation interface observed in the crystal packing of both human Mis18$_{Yippee}$ and *S. pombe* Mis18$_{Yippee}$ (Fig. EV3C, highlighted by zoom in view). Accordingly, disrupting this interface by mutating Mis18α residues C154 and D160 (Fig. EV3C) perturbed Mis18 oligomerisation as evidenced by SEC analysis (Fig. EV3D).

## Mis18α oligomerisation via the C-terminal helical bundle assembly is essential for Mis18α/β centromere localisation and new CENP-A loading

Although the subunit stoichiometry and the arrangement of Mis18α/β C-terminal helices within the helical bundle proposed by Nardi et al, 2016 are different from the data presented here, the Mis18α residues (I201, L205, L212, L215 and L219) that were predicted by them to stabilise the helical bundle do indeed form the 'spine' of the hydrophobic core running along the triple-helical bundle (Fig. 1E,F). Mutating these residues perturbed the ability of Mis18α tethered at an ectopic LacO site to facilitate CENP-A deposition at the tethering site (Nardi et al, 2016). However, how these Mis18α mutants perturb the oligomeric structure of the Mis18α/β C-terminal helical bundle and how this structural perturbation affects CENP-A loading at endogenous centromeres remain as open questions.

To address these questions, we first tested these mutants using in vitro amylose pull-down assays by mixing recombinantly purified WT and mutant His-MBP-Mis18β$_{188-229}$ and His-SUMO-Mis18α$_{191-233}$ proteins. Mutating these residues to Ala

(Mis18α$_{I201A/L205A}$ and Mis18α$_{L212A/L215A/L219A}$) or Asp (Mis18α$_{I201D/L205D}$) abolished the ability of Mis18α α-helix to interact with Mis18β$_{188-229}$ (Fig. EV4A). Co-immunoprecipitation (Co-IP) assays using an anti-Mis18α antibody were performed on cells where endogenous Mis18α was depleted, and Mis18α-mCherry was co-expressed with Mis18β-GFP to check for complex formation (Fig. EV4B). In line with our in vitro pull-downs, the Co-IPs using a Mis18α antibody revealed that Mis18α$_{WT}$-mCherry interacted with Mis18β-GFP while Mis18α$_{I201A/L205A}$ and Mis18α$_{L212A/L215A/L219A}$ mutants did not (Fig. EV4B). SEC-MALS analysis of His-SUMO tagged Mis18α$_{188-233}$ showed that on its own, Mis18α WT protein can form a dimer, whilst introducing I201A/L205A or L212A/L215A/L219A results in both proteins forming a monomer (Fig. EV4C). To evaluate the role of C-terminal helical bundle assembly, mediated via the Mis18α oligomerisation, on centromere localisation of Mis18α and Mis18β and CENP-A deposition, these mutants were further tested in HeLa cells.

HeLa Mis18β-GFP CENP-A-SNAP cells (McKinley and Cheeseman, 2014) were depleted of endogenous Mis18α by siRNA (Fig. EV4D) and simultaneously rescued with either WT or mutant Mis18α-mCherry (Fig. EV4E), then visualised by immunofluorescence along with ACA. Unlike Mis18α$_{WT}$, the Mis18α mutants (Mis18α$_{I201A/L205A}$, Mis18α$_{I201D/L205D}$ and Mis18α$_{L212A/L215A/L219A}$) all failed to localise to centromeres (Fig. 3A). As expected, Mis18β-GFP co-expression showed co-localisation between Mis18β$_{WT}$ with Mis18α$_{WT}$. However, in cells expressing Mis18α$_{I20A1/L205A}$, Mis18α$_{I201D/L205D}$ and Mis18α$_{L212A/L215A/L219A}$, Mis18β could no longer co-localise with Mis18α at the centromere. Together, this confirms that Mis18β depends on its interaction with Mis18α and the formation of the C-terminal triple helical assembly to localise at centromeres.

We then evaluated the impact of Mis18α mutants not capable of forming the C-terminal helical bundle on new CENP-A deposition. We did this by performing a Quench-Chase-Pulse CENP-A-SNAP Assay according to Jansen et al (Jansen et al, 2007) (Fig. 3B). HeLa CENP-A-SNAP cells were depleted of endogenous Mis18α and rescued with either Mis18α$_{WT}$ or Mis18α mutants (Mis18α$_{I20A1/L205A}$, Mis18α$_{I201D/L205D}$ and Mis18α$_{L212A/L215A/L219A}$). The existing CENP-A was blocked with a non-fluorescent substrate of the SNAP, and the new CENP-A deposition in the early G1 phase was visualised by staining with the fluorescent substrate of the SNAP. Mis18α$_{WT}$ rescued new CENP-A deposition to levels compared to that of control siRNA (Fig. 3C). However, Mis18α$_{I20A1/L205A}$, Mis18α$_{I201D/L205D}$ and Mis18α$_{L212A/L215A/L219A}$ abolished new CENP-A loading almost completely, indicating that the formation of the Mis18 triple-helical bundle is essential for CENP-A deposition (Fig. 3C).

**A**

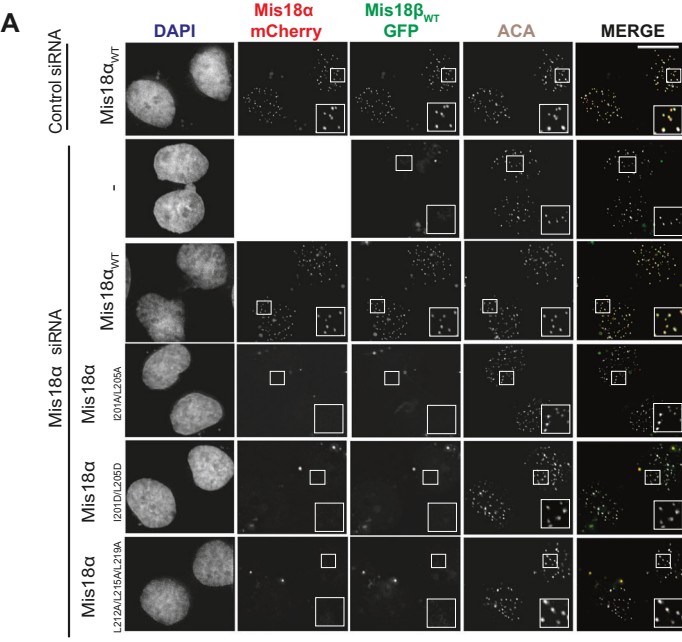

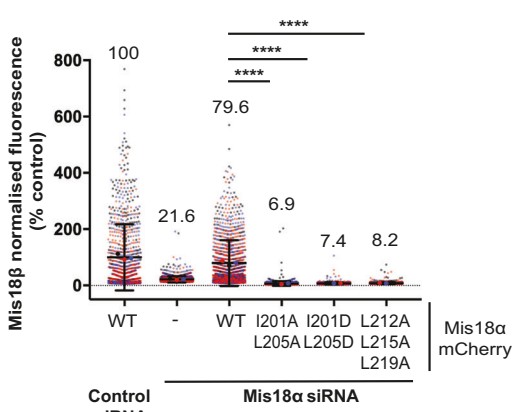

**B**

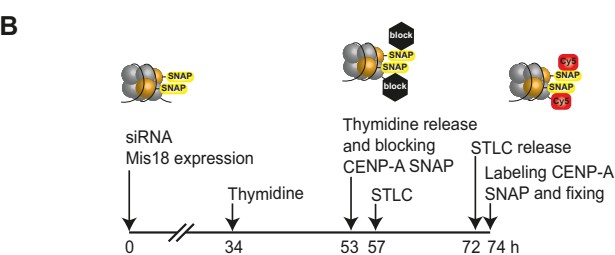

**C**

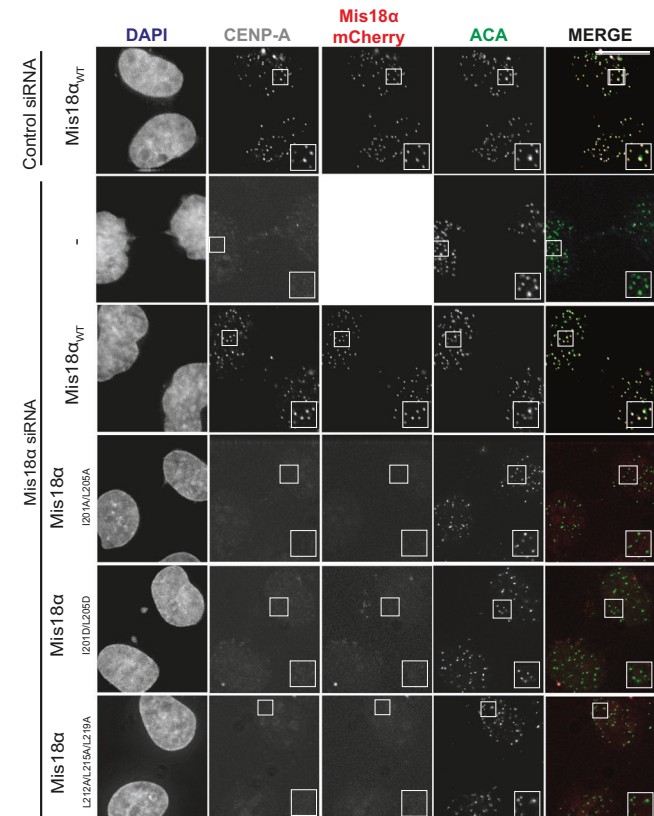

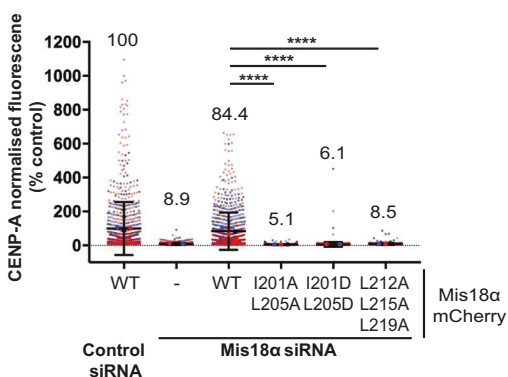

Figure 3. Mis18α mutations disrupting the Mis18α/β triple helical assembly result in loss of Mis18α/β centromere localisation and CENP-A deposition.

## Mis18α associates with the centromere independently of Mis18β and can deposit CENP-A, but efficient CENP-A loading requires Mis18β

We again performed amylose in vitro pull-down assays, using His-SUMO-Mis18α$_{191-233\ WT}$ and mutant His-MBP-Mis18β$_{188-229}$ proteins, to assess the ability of Mis18β mutant to form a triple-helical bundle with Mis18α. Based on our X-ray crystal structure (Fig. 1E), we identified one cluster (L199/I203) in Mis18β and observed that mutating these residues to either Ala (Mis18β$_{L199A/I203A}$) or Asp (Mis18β$_{L199D/I203D}$) either reduced or abolished its ability to interact with Mis18α$_{191-233}$ (Fig. 4A, left panel). Co-IP analysis using an anti-Mis18α antibody was performed on cells where endogenous Mis18β was depleted, and Mis18β-GFP was expressed along Mis18α-mCherry to check for complex formation. Western blot analysis showed that Mis18β$_{WT}$ could interact with Mis18α-mCherry and that the ability of Mis18β$_{L199D/I203D}$ to interact with Mis18α was reduced (Fig. 4A, right panel).

To assess the contribution of Mis18β for the centromere association and function of Mis18α, we evaluated the Mis18β mutant (Mis18β$_{L199D/I203D}$), which cannot form the triple-helical assembly with Mis18α, in siRNA rescue assays by expressing Mis18β-GFP-tagged proteins in a mCherry-Mis18α cell line (McKinley and Cheeseman, 2014). Depletion of endogenous Mis18β and simultaneous transient expression of Mis18β$_{WT}$-GFP led to co-localisation of Mis18β with Mis18α at centromeres (Figs. 4B and EV4D,E). Under these conditions, Mis18β$_{WT}$-GFP levels at centromeres were comparable to that of the control siRNA. Whereas Mis18β$_{L199D/I203D}$ failed to localise at the centromeres. Strikingly, Mis18β$_{L199D/I203D}$ perturbed centromere association of Mis18α only moderately (Fig. 4B). This suggests that Mis18α can associate with centromeres in a Mis18β independent manner.

Next, we assessed the contribution of Mis18β for CENP-A deposition in the Quench-Chase-Pulse CENP-A-SNAP assay described above. Endogenous Mis18β was depleted using siRNA, and Mis18β$_{WT}$ and Mis18β$_{L199D/I203D}$ were transiently expressed as GFP-tagged proteins in HeLa cells expressing CENP-A-SNAP. Mis18β$_{WT}$ rescued new CENP-A deposition to comparable levels to the ones observed in the control siRNA-Mis18β WT condition (Fig. 4C). Interestingly, unlike the Mis18α mutants (Mis18α$_{I20A1/L205A}$, Mis18α$_{I201D/L205D}$ and Mis18α$_{L212A/L215A/L219A}$), Mis18β$_{L199D/I203D}$ did not abolish new CENP-A loading but reduced the levels only moderately.

Together, these analyses demonstrate that Mis18α can associate with centromeres and deposit new CENP-A independently of Mis18β. However, efficient CENP-A loading requires Mis18β.

## Structural basis for centromere recruitment of Mis18α/β by Mis18BP1

Previous studies have shown that the first 130 amino acids of Mis18BP1 are required to bind Mis18α/β$_{Yippee}$ domains (Spiller et al, 2017). However, how Mis18α/β$_{Yippee}$ domains recognise Mis18BP1 is not clear. Our structural analysis based on the AlphaFold model suggests that two Mis18BP1 fragments, a short helical segment spanning aa residues 109–130 (Mis18BP1$_{109-130}$) and a region spanning aa residues 20–51 (Mis18BP1$_{20-51}$) interact with Mis18α$_{Yippee}$ domain and with an interface formed between Mis18α/β$_{Yippee}$ heterodimers, respectively (Fig. 5A). Mis18BP1$_{109-130}$ binds at a hydrophobic pocket of the Mis18α$_{Yippee}$ domain formed by amino acids L83, F85, W100, I110, V172 and I175. This hydrophobic pocket is surrounded by hydrophilic amino acids E103, D104, T105, S169 E171 facilitating additional electro-static interactions with Mis18BP1$_{109-130}$ (Fig. 5A). Mis18BP1$_{20-51}$ contains two short β strands that interact at Mis18α/β$_{Yippee}$ interface extending the six-stranded-β sheets of both Mis18α and Mis18β Yippee domains. This provides the structural rationale for why Yippee domains-mediated Mis18α/β hetero-hexamerisation is crucial for Mis18BP1 binding (Spiller et al, 2017). Notably, the two Cdk1 phosphorylation sites on Mis18BP1 (T40 and S110) that we and others have shown to disrupt Mis18 complex assembly (Pan et al, 2017; Spiller et al, 2017) lie directly within the Mis18α/β binding interface predicted by this model providing the structural basis for Cdk1 mediated regulation of Mis18 complex assembly. Consistent with this model, several cross-links observed between Mis18BP1 and Mis18α and Mis18β map to these residues. Mutating the negatively charged amino acid cluster of Mis18α (E103, D104 and T105) that is juxtaposed to Mis18BP1$_{109-130}$ in a TetR-eYFP-Mis18α vector (TetR-eYFP-Mis18α$_{E103R/D104R/T105R}$) transfected in HeLa cells with an ectopic synthetic alphoid$^{tetO}$ array integrated in a chromosome arm significantly perturbed Mis18α's ability to recruit Mis18BP1$_{20-130}$-mCherry to the tethering site as compared to Mis18α$_{WT}$ (Fig. 5B).

Furthermore, we probed the effects of perturbing Mis18α-Mis18BP1 interaction on endogenous centromeres. We depleted Mis18α in a cell line that stably expresses CENP-A-SNAP and allows inducible expression of GFP-Mis18BP1 (McKinley and Cheeseman, 2014). We then assessed the ability of transfected Mis18α-mCherry to co-localise with Mis18BP1 at centromeres. Depletion of Mis18α and simultaneous expression of either Mis18α$_{WT}$-mCherry or Mis18α$_{E103R/D104R/T105A}$-mCherry revealed that, unlike Mis18α$_{WT}$, Mis18α$_{E103R/D104R/T105A}$ failed to localise at endogenous centromeres (Fig. 5C, middle panel). We also observed

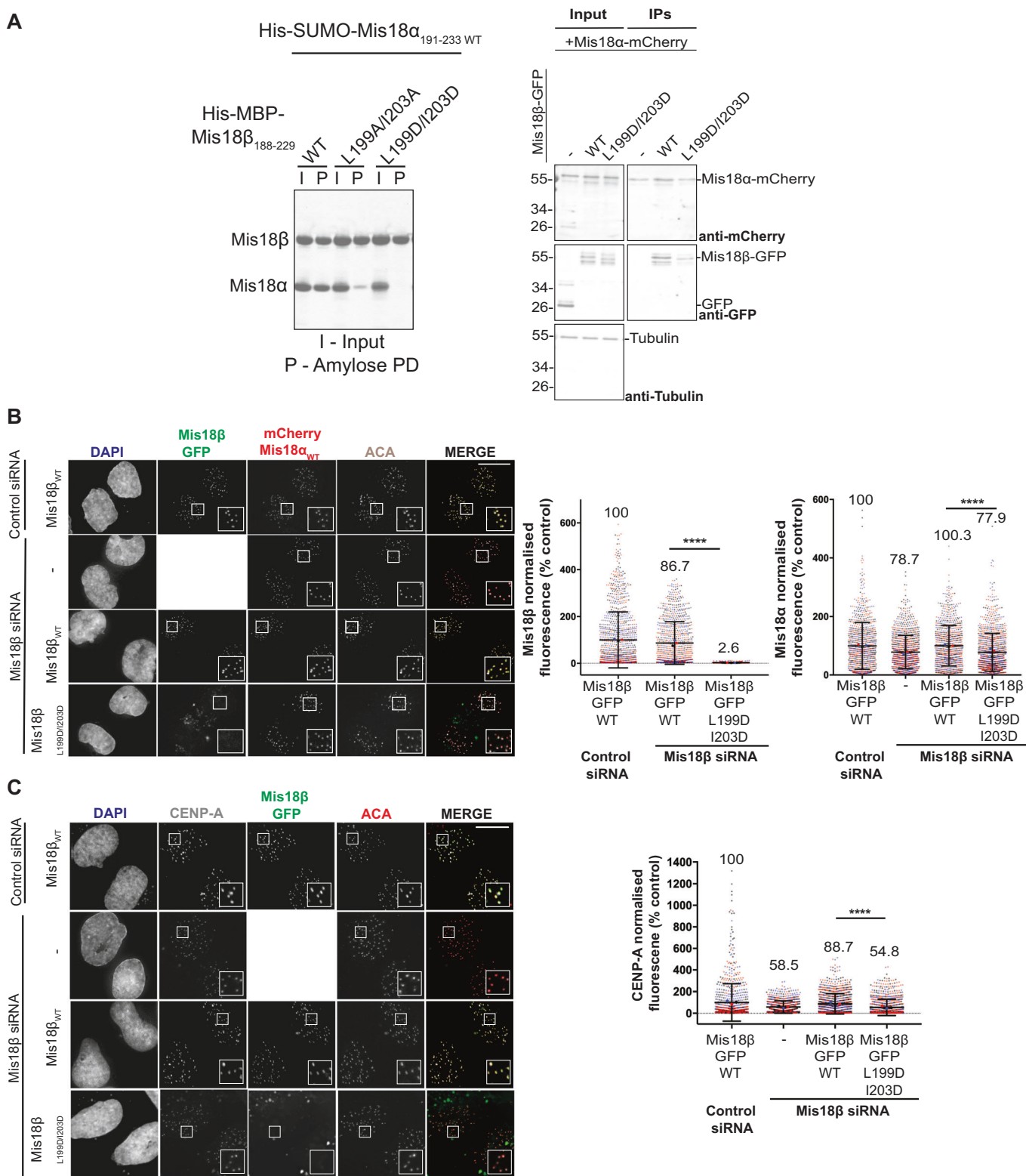

a slight decrease in the levels of GFP-Mis18BP1 at the centromere when Mis18α$_{E103R/D104R/T105A}$ was expressed as compared to Mis18α$_{WT}$ (Fig. 5C, right panel). Consistent with the observation of reduced centromeric Mis18α, when Mis18α$_{E103R/D104R/T105A}$-

mCherry is expressed, the quantification of new CENP-A deposition in HeLa cell expressing CENP-A-SNAP showed a significant reduction of new CENP-A deposition at the centromere indicating that the interaction of Mis18α with Mis18BP1 is essential for

◄ **Figure 4.   Mis18α associates with centromeres in a Mis18β-independent manner but requires Mis18β for efficient CENP-A loading.**

(A) Left panel shows SDS-PAGE analysis of cobalt and amylose pull-down of His-MBP-Mis18β$_{188-229}$ WT and mutants with His-SUMO-Mis18α$_{191-233}$. SDS-PAGE shows protein bound to nickel resin as input (I) and protein bound to amylose resin to assess interaction (P). Right panel shows Western blot analysis of co-immunoprecipitation (Co-IP) experiments using Mis18α antibody to test interaction of Mis18α–mCherry and Mis18β-GFP with and without mutations in the C-terminal α-helices or GFP as a control. Top panel shows blot against mCherry, middle panel shows blot against GFP, and bottom panel shows blot against tubulin as loading control. (B) Representative fluorescence images (left panel) and quantification (right panel) used to evaluate the ability of Mis18β$_{WT}$-GFP ($n = 963, 927$) and Mis18β$_{L199D/I203D}$-GFP ($n = 1312, 1221$) to co-localise with mCherry-Mis18α at endogenous centromeres. Middle panel, quantification of Mis18β signal. Right panel, quantification of Mis18α signal (Mann–Whitney $U$ test; ****$P \leq 0.0001$). Cells were co-transfected with either control ($n = 1131, 935$) or Mis18β siRNA (with no transfected Mis18β-GFP $n = 1170$), in three independent experiments shown in black, blue and red. Error bars show mean ± SD. (C) Representative fluorescence images (left panel) and quantification (right panel) used to evaluate the ability of Mis18β$_{WT}$-GFP ($n = 1036$) and Mis18β$_{L199D/I203D}$ GFP ($n = 947$) to deposit new CENP-A-SNAP at endogenous centromeres (Mann–Whitney $U$ test; ****$P \leq 0.0001$). Cells were co-transfected with either control ($n = 840$) or Mis18β siRNA (with no transfected Mis18β-GFP $n = 824$), as stated, in three independent experiments shown in black, blue and red. Error bars show mean ± SD. Scale bars, 10 µm. All conditions have been normalised to control conditions: cells transfected with control siRNA and Mis18β$_{WT}$-GFP. Source data are available online for this figure.

centromeric recruitment of the Mis18 complex and for CENP-A loading (Fig. 5D).

## Discussion

Mis18 complex assembly is a central process essential for the recruitment of CENP-A/H4 bound HJURP and the subsequent CENP-A deposition at centromeres (Dunleavy et al, 2009; Fujita et al, 2007; Jansen et al, 2007). Thus far, several studies, predominantly biochemical and cellular, have characterised inter-actions and functions mediated by the two distinct structural domains of the Mis18 proteins, the Yippee and C-terminal α-helical domains of Mis18α and Mis18β (Nardi et al, 2016; Pan et al, 2017; Spiller et al, 2017; Stellfox et al, 2016). Some of the key conclusions of these studies include: (1) Mis18α/β is a hetero-hexamer made of 4 Mis18α and 2 Mis18β; (2) The Yippee domains and C-terminal α-helices of Mis18α and Mis18β have the intrinsic ability to homo- or hetero-oligomerise, and form three distinct oligomeric modules in different copy numbers—a Mis18α$_{Yippee}$ homodimer, two copies of Mis18α/β$_{Yippee}$ heterodimers and two heterotrimers made of Mis18α/β C-terminal helices (2 Mis18α and 1 Mis18β); (3) the two copies of Mis18α/β$_{Yippee}$ heterodimers each bind one Mis18BP1$_{20-130}$ and form a hetero-octameric Mis18$_{core}$ complex (Mis18α/Mis18β/Mis18BP1$_{20-130}$: a Mis18α/β hetero-hexamer bound to 2 copies of Mis18BP1$_{20-130}$). However, no experimentally determined structural information is available for the human Mis18 complex. This is crucial to identify the amino acid residues essential for the assembly of Mis18α/β and the holo-Mis18 complexes and to determine the specific interactions that are essential for the localisation of Mis18 complex to centromeres and its function.

Here, we have taken an integrative structural approach that combines X-ray crystallography, electron microscopy and homology modelling with cross-linking mass spectrometry to characterise the structure of the Mis18 complex. Our analysis shows that Mis18α/β heterotrimer is stabilised by the formation of a triple-helical bundle with a Mis18α/β$_{Yippee}$ heterodimer and Mis18α$_{Yippee}$ monomer arranged as a linear array. Two such Mis18α/β heterotrimers assemble as a hetero-hexamer via the homodimer-isation of the Mis18α$_{Yippee}$ domains. The crystal structure of Mis18α/β$_{C-term}$ triple-helical structure allowed us to design several separation-of-function Mis18α and Mis18β mutants. These muta-tions specifically perturb the ability of Mis18α or Mis18β to assemble into the helical bundle, while retaining their other

functions, if there are any. Functional evaluation of these mutants in cells has provided important new insights into the molecular interdependencies of the Mis18 complex subunits. Particularly, the observations that: (1) Mis18α can associate with centromeres and deposit CENP-A independently of Mis18β, and (2) depletion of Mis18β or disrupting the incorporation of Mis18β into the Mis18 complex, while does not abolish CENP-A loading, reduces the CENP-A deposition amounts, questions the consensus view that Mis18α and Mis18β always function as a single structural entity to exert their function to maintain centromere maintenance.

Whilst proteins involved in CENP-A loading have been well-established, the mechanism by which the correct levels of CENP-A are controlled is yet to be thoroughly explored and characterised. The data presented here suggest that Mis18β mainly contributes to the quantitative control of centromere maintenance by ensuring the right amount of CENP-A deposition at centromeres. We also note that the Mis18β mutant, which cannot interact with Mis18α, moderately reduced Mis18α levels at centromeres, and hence, it is possible that Mis18β ensures the correct level of CENP-A deposition by facilitating optimal Mis18α centromere recruitment. Future studies will focus on dissecting the mechanisms underlying the Mis18β-mediated control of CENP-A loading amounts along with any other mechanisms involved.

Previously published work identified amino acid sequence similarity between the N-terminal region of Mis18α and R1 and R2 repeats of the HJURP that mediates Mis18α/β interaction (Pan et al, 2019). Deletion of the Mis18α N-terminal region enhanced HJURP interaction with the Mis18 complex (Pan et al, 2019). Here, we show that the N-terminal helical region of Mis18α makes extensive contact with the C-terminal helices of Mis18α and Mis18β, which had previously been shown to mediate HJURP binding by Pan et al, 2019. Collectively these observations suggest that the N-terminal region of Mis18α might directly interfere with HJURP– Mis18 complex interaction. Two independent recent studies (preprint: Conti et al, 2024; preprint: Parashara et al, 2024) reveal that this is indeed the case and a Plk1-mediated phosphorylation cascade involving several phosphorylation and binding events of the Mis18 complex subunits relieve the intramolecular interactions between the Mis18α N-terminal helical region and the HJURP binding surface of the Mis18α/β C-terminal helical bundle. This facilitates robust HJURP–Mis18α/β interaction in vitro and efficient HJURP centromere recruitment and CENP-A loading in cells. Overall, these studies also highlight the importance of the critical structural insights into the Mis18 complex we report here.

One of the key outstanding questions in the field is how does the Mis18 complex associate with the centromere. Previous studies identified

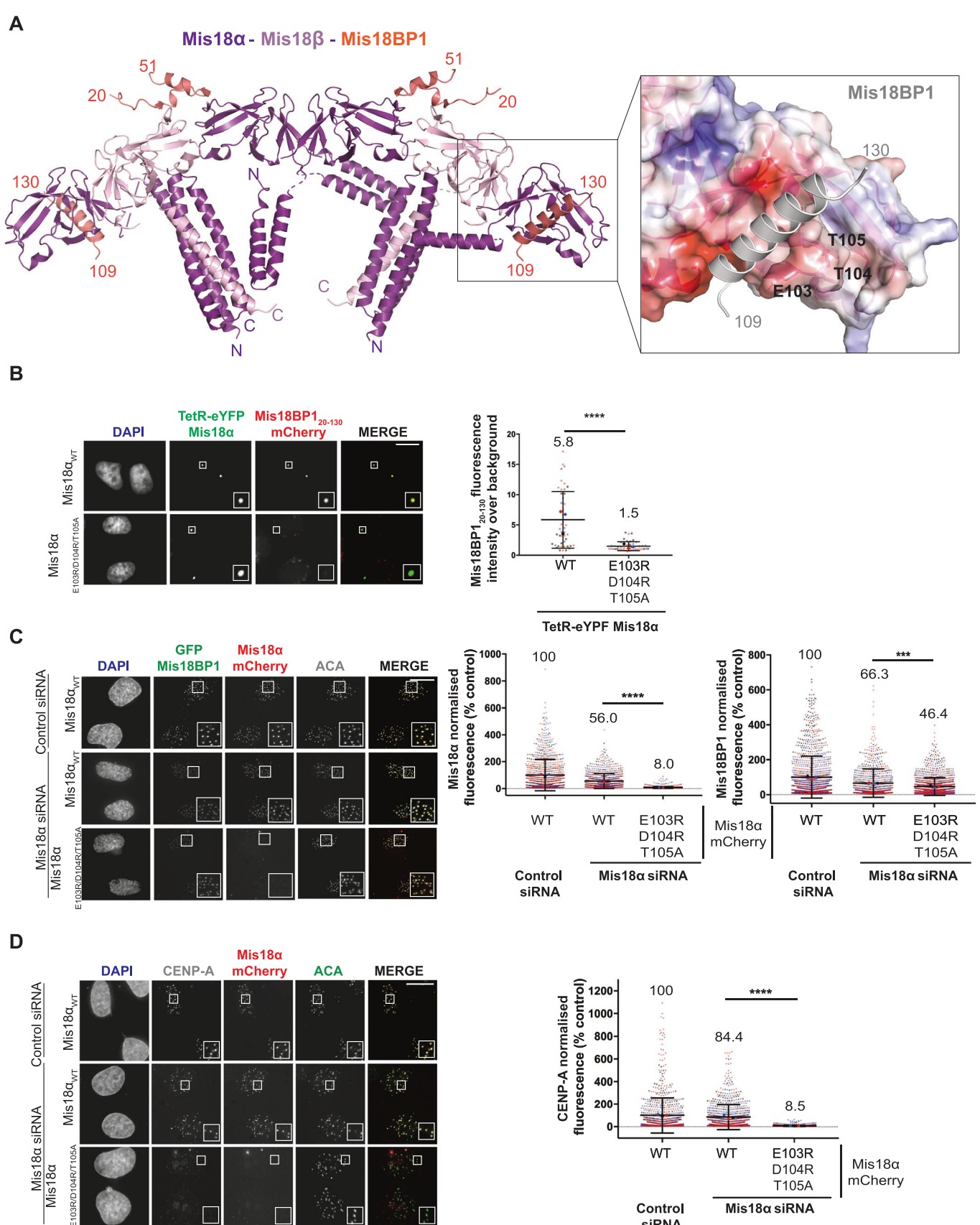

**Figure 5.** Disrupting the Mis18BP1 binding interface of Mis18α prevents its centromere localisation and CENP-A deposition.

(A) Mis18α/Mis18β model and its surface representation coloured based on electrostatic surface potential (zoom panel), highlighting the residues proposed to be involved in Mis18BP1 binding. Mis18α shown in purple, Mis18β shown in light pink and Mis18BP1 shown in salmon or grey for the zoom panel for clarity. (B) Representative images and quantification showing the recruitment of either Mis18BP1$_{20-130}$-mCherry by different Mis18α constructs (WT and mutant) tethered to the alphoid$^{tetO}$ array in HeLa 3–8. Tethering of TetR-eYFP-Mis18α$_{WT}$ ($n = 45$) and TetR-eYFP-Mis18α$_{E103R/D104R/T105A}$ ($n = 46$) testing recruitment of Mis18BP1$_{20-130}$ mCherry (Mann–Whitney $U$ test; ****$P \leq 0.0001$). Data from three independent experiments shown in black, blue and red. Error bars show mean ± SD. Scale bars, 10 μm. (C) Representative fluorescence images (left panel) and quantifications (right panel) evaluating the ability of Mis18α$_{WT}$-mCherry ($n = 985, 856$) and Mis18α$_{E103R/D104R/T105A}$ ($n = 1497, 1511$) to co-localise with GFP-Mis18BP1 at endogenous centromeres. Middle panel, quantification of Mis18α signal and right panel, quantification of Mis18BP1 signal (Mann–Whitney $U$ test; ***$P \leq 0.001$, ****$P \leq 0.0001$). Cells were co-transfected with either control ($n = 1016, 1403$) or Mis18α siRNA, as stated, in three independent experiments shown in black, blue and red. Error bars show mean ± SD. Scale bars, 10 μm. (D) Representative fluorescence images (left panel) and quantifications (right panel) evaluating the ability of Mis18α$_{WT}$-mCherry ($n = 896$) and Mis18α$_{E103R/D104R/T105A}$ ($n = 1430$) to deposit new CENP-A-SNAP at endogenous centromeres (Mann–Whitney $U$ test; ****$P \leq 0.0001$). Cells were co-transfected with either control ($n = 852$) or Mis18α siRNA, as stated, in three independent experiments shown in black, blue and red. Error bars show mean ± SD. Scale bars, 10 μm. All conditions have been normalised to control conditions: cells transfected with control siRNA and Mis18α$_{WT}$-mCherry. Source data are available online for this figure.

CCAN subunits CENP-C and CENP-I as major players mediating the centromere localisation of the Mis18 complex mainly via Mis18BP1 (Dambacher et al, 2012; Moree et al, 2011; Shono et al, 2015), although Mis18β subunit has also been suggested to interact with CENP-C (Stellfox et al, 2016). Within the Mis18 complex, we and others have shown that the Mis18α/β$_{Yippee}$ heterodimers can directly interact with Mis18BP1. Here our structural analysis allowed us to map the interaction interface mediating Mis18α/β-Mis18BP1 binding. Perturbing this interface on Mis18α completely abolished Mis18α centromere localisation and reduced Mis18BP1 centromere levels. These observations show that Mis18α associates with the centromere mainly via Mis18BP1, and assembly of the Mis18 complex itself is crucial for its efficient centromere association, as previously suggested. Future work aimed at characterising the intermolecular contact points between the subunits of the Mis18 complex, centromeric chromatin and CCAN components and understanding if the Mis18 complex undergoes any conformational and/or compositional variations upon centromere association and/or during CENP-A deposition process, will be crucial to delineate the mechanisms underpinning the centromere maintenance.

# Methods

## Plasmids

For crystallisation, a polycistronic expression vector for the C-terminal coiled-coil domains of Mis18α (residues 191–233, Mis18α$_{C-term}$) and Mis18β (residues 188–229, Mis18β$_{C-term}$) were produced with the N-terminal 6His-SUMO- (His-SUMO) and 6His-MBP-tags (His-MBP), respectively. Mis18α$_{Yippee}$ (residues 77–190) was cloned into the pET3a vector with the N-terminal 6His-tag.

For all other recombinant proteins, codon-optimised sequences (GeneArt) for Mis18α and Mis18β were cloned into pET His6 TEV or pET His6 msfGFP TEV (9B Addgene plasmid #48284, 9GFP Addgene plasmid #48287, a kind gift from Scott Gradia), respectively. They were combined to make a single polycistronic plasmid. The boundaries of ΔN for Mis18α and Mis18β were 77–187 and 56–183 Mis18BP1$_{20-130}$ was cloned in pEC-K-3C-His-GST and pET His6 MBP TEV (9C Addgene plasmid #48286).

Non-codon-optimised sequences were amplified from a human cDNA library (MegaMan human transcription library, Agilent). Mis18α, Mis18β and Mis18BP1$_{20-130}$ were cloned into pcDNA3 mCherry LIC vector, pcDNA3 GFP LIC vector (6B Addgene

plasmid #30125, 6D Addgene plasmid #30127, a kind gift from Scott Gradia) and TetR-eYFP-IRES-Puro vector as stated. All mutations were generated following QuikChange site-directed mutagenesis protocol (Stratagene), using primers in Table EV3.

## Expression and purification of recombinant proteins

For crystallisation, both Mis18α/β$_{C-term}$ domains and Mis18α$_{Yippee}$ were transformed and expressed in *Escherichia coli* BL21 (DE3) using the auto-inducible expression system (Studier, 2005). The cells were harvested and resuspended in the lysis buffer containing 30 mM Tris-HCl pH 7.5, 500 mM NaCl and 5 mM β-mercaptoethanol with protease inhibitor cocktails. The resuspended cells were lysed using the ultra-sonication method and centrifuged at $20,000 \times g$ for 50 min at 4 °C to remove the cell debris. After 0.45 μm filtration of the supernatant, the lysate was loaded into the cobalt affinity column (New England Biolabs) and eluted with a buffer containing 30 mM Tris-HCl pH 7.5, 500 mM NaCl, 5 mM β-mercaptoethanol, and 300 mM imidazole. The eluate was loaded into the amylose affinity column (New England Biolabs) and washed with a buffer containing 30 mM Tris-HCl pH 7.5, 500 mM NaCl and 5 mM β-mercaptoethanol. To cleave the His-MBP tag, on-column cleavage was performed by adding Tobacco Etch Virus (TEV) protease (1:100 ratio) into the resuspended amylose resin and incubated overnight at 4 °C. The TEV cleavage released the untagged Mis18α/β$_{C-term}$ domains in solution, and the flow-through fraction was collected and concentrated using a Centricon (Millipore). The protein was loaded onto a HiLoad™ 16/600 Superdex™ 200 column (GE Healthcare) equilibrated with a buffer containing 30 mM Tris-HCl pH 7.5, 100 mM NaCl and 1 mM TCEP. To further remove the contaminated MBP tag, the sample was re-applied into the amylose affinity column, and the flow-through fraction was collected and concentrated to 20 mg/ml for the crystallisation trial. SeMet (selenomethionine) incorporated Mis18α/β$_{C-term}$ domains were expressed with PASM-5052 auto-inducible media (Studier, 2005). The SeMet-substituted Mis18α/β$_{C-term}$ domains were purified using the same procedure described above.

The purification of His tagged Mis18α$_{Yippee}$ employed the same purification method used for Mis18α/β$_{C-term}$ domains except for the amylose affinity chromatography step. The purified Mis18α$_{Yippee}$ from the HiLoad™ 16/600 Superdex™ 200 chromatography was concentrated to 13.7 mg/ml with the buffer containing 30 mM Tris-HCl pH 7.5, 100 mM NaCl and 1 mM TCEP.

All other proteins were expressed in *Escherichia coli* BL21 (*DE3*) Gold cells using LB. After reaching an O.D. ~0.6 at 37 °C, cultures were cooled to 18 °C and induced with 0.35 mM IPTG overnight. The His-Mis18α/His-GFP-Mis18β complex was purified by resuspending the pellet in a lysis buffer containing 20 mM Tris-HCl pH 8.0 at 4 °C, 250 mM NaCl, 35 mM imidazole pH 8.0 and 2 mM β-mercaptoethanol supplemented with 10 µg/ml DNase, 1 mM PMSF and cOmplete™ EDTA-free (Sigma). After sonication, clarified lysates were applied to a 5 ml HisTrap™ HP column (GE Healthcare) and washed with lysis buffer followed by a buffer containing 20 mM Tris-HCl pH 8.0 at 4 °C, 1 M NaCl, 35 mM imidazole pH 8.0, 50 mM KCl, 10 mM MgCl₂, 2 mM ATP and 2 mM β-mercaptoethanol and then finally washed with lysis buffer. The complex was then eluted with 20 mM Tris-HCl pH 8.0 at 4 °C, 250 mM NaCl, 500 mM imidazole pH 8.0 and 2 mM β-mercaptoethanol. Fractions containing proteins were pooled, and TEV was added (if needed) whilst performing overnight dialyses against 20 mM Tris-HCl pH 8.0 at 4 °C, 150 mM NaCl and 2 mM DTT.

His-GST-Mis18BP1$_{20-130}$ was purified in the same manner as above with the following modifications: the lysis and elution buffers contained 500 mM NaCl, whilst the dialysis buffer contained 75 mM NaCl. His-MBP-Mis18BP1$_{20-130}$ was purified using the same lysis buffer containing 500 mM NaCl and purified using amylose resin (New England Biolabs). Proteins were then eluted by an elution buffer containing 10 mM maltose.

If needed, proteins were subjected to anion exchange chromatography using the HiTrap™ Q column (GE Healthcare) using the ÄKTA™ start system (GE Healthcare). Concentrated fractions were then injected onto either Superdex™ 75 increase 10/300 or Superdex™ 200 increase 10/300 columns equilibrated with 20 mM Tris-HCl pH 8.0 at 4 °C, 100–250 mM NaCl and 2 mM DTT using the ÄKTA™ Pure 25 system (GE Healthcare).

## Interaction trials

Pull-down assays used to test the interaction between the C-terminus of Mis18α and Mis1β were performed by initially purifying the proteins through the cobalt affinity chromatography, as described for wild-type proteins, and the eluted fractions were loaded into the amylose affinity resin, pre-equilibrated with a binding buffer consisting of 30 mM Tris-HCl pH 7.5, 500 mM NaCl and 5 mM β-mercaptoethanol. Amylose resins were washed with the binding buffer, and the proteins were eluted with a binding buffer containing 20 mM maltose. The fractions were subjected to SDS-PAGE analysis.

Pull-down assay using the amylose resin to test interactions between Mis18α/β and Mis18BP1$_{20-130}$ were done as described previously (Pan et al, 2017). Briefly, purified proteins were diluted to 10 µM in 40 µl binding buffer, 50 mM HEPES pH 7.5, 1 M NaCl, 1 mM TCEP, 0.01% Tween® 20. One-third of the mixture was taken as input, and the remaining fraction was incubated with 40 µl amylose resin for 1 h at 4 °C. The bound protein was separated by washing with binding buffer three times, and the input and bound fractions were analysed by SDS-PAGE.

## Crystallisation, data collection, and structure determination

Purified Mis18α/β$_{C-term}$ domains and Mis18α$_{Yippee}$ were screened and crystallised using the hanging-drop vapour diffusion method at

room temperature with a mixture of 0.2 µl of the protein and 0.2 µl of crystallisation screening solutions. The crystals of Mis18α/β$_{C-term}$ domains were grown within a week with a solution containing 0.2 M magnesium acetate and 20% (w/v) PEG 3350. SeMet-substituted Mis18α/β$_{C-term}$ domains crystals were grown by the micro-seeding method with a solution containing 0.025 M magnesium acetate and 14% (w/v) PEG 3350. The crystals of SeMet-substituted Mis18α/β$_{C-term}$ domains were further optimised by mixing 1 µl of the protein and 1 µl of the optimised crystallisation solution containing 0.15 M magnesium acetate and 20% (w/v) PEG 3350. The crystals of Mis18α$_{Yippee}$ were obtained in 2 M ammonium sulfate, 2% (w/v) PEG 400, and 100 mM HEPES at pH 7.5. The crystals of Mis18α/β$_{C-term}$ domains and Mis18α$_{Yippee}$ were cryoprotected with the crystallisation solutions containing 20% and 25% glycerol, respectively. The cryoprotected crystals were flash-frozen in liquid nitrogen. Diffraction datasets were collected at the beamline LS-CAT 21 ID-G and ID-D of Advanced Photon Source (Chicago, USA). The data set were processed and scaled using the DIALS (Winter et al, 2018) via Xia2 (Winter et al, 2013). The initial model of Mis18α/β$_{C-term}$ domains was obtained using the SAD method with SeMet-derived data using the Autosol program (Terwilliger, 2000). The molecular replacement of the initial model as a search model against native diffraction data was performed using the Phaser programme within the PHENIX programme suite (Liebschner et al, 2019). The initial model of Mis18α$_{Yippee}$ was calculated by molecular replacement method (Phaser) using yeast Mis18 Yippee-like domain structure (PDB ID: 5HJ0) (Subramanian et al, 2016) as a search model. The final structures were manually fitted using the Coot programme (Emsley and Cowtan, 2004) and the refinement was carried out using REFMAC5 (Afonine et al, 2010). The quality of the final structures was validated with the MolProbity programme (Chen et al, 2010).

## SEC-MALS

Size-exclusion chromatography (ÄKTA-MicroTM, GE Healthcare) coupled to UV, static light scattering and refractive index detection (Viscotek SEC-MALS 20 and Viscotek RI Detector VE3580; Malvern Instruments) was used to determine the molecular mass of protein and protein complexes in solution. Injections of 100 µl of 2–6 mg/ml material were used.

His-SUMO-Mis18α$_{188-233}$ ($\partial A_{280nm}/\partial c = 0.43$ AU.ml.mg⁻¹) WT and mutants were run on a Superdex 75 increase 10/300 GL size-exclusion column pre-equilibrated in 50 mM HEPES pH 8.0, 150 mM NaCl and 1 mM TCEP at 22 °C with a flow rate of 1.0 ml/min. Light scattering, refractive index (RI) and A$_{280nm}$ were analysed by a homo-polymer model (OmniSEC software, v5.02; Malvern Instruments) using the parameters stated for the protein, $\partial n/\partial c = 0.185$ ml.g⁻¹ and buffer RI value of 1.335. The mean standard error in the mass accuracy determined for a range of protein-protein complexes spanning the mass range of 6–600 kDa is ±1.9%.

## SAXS

SEC-SAXS experiments were performed at beamline B21 of the Diamond Light Source synchrotron facility (Oxfordshire, UK). Protein samples at concentrations >5 mg/ml were loaded onto a Superdex™ 200 Increase 10/300 GL size-exclusion chromatography column (GE

Healthcare) in 20 mM Tris pH 8.0, 150 mM KCl at 0.5 ml/min using an Agilent 1200 HPLC system. The column outlet was fed into the experimental cell, and SAXS data were recorded at 12.4 keV, detector distance 4.014 m, in 3.0 s frames. Data were subtracted, averaged and analysed for Guinier region $Rg$ and cross-sectional $Rg$ ($Rc$) using ScÅtter 3.0 (ScÅtter), and $P(r)$ distributions were fitted using PRIMUS (Konarev et al, 2003). Ab initio modelling was performed using DAMMIN (Svergun, 1999), in which 30 independent runs were performed in P1 or P2 symmetry and averaged.

## Gradient fixation (GraFix)

Fractions from the gel filtration peak were concentrated to 1 mg/mL using a Vivaspin® Turbo (Sartorius) centrifugal filter, and the buffer exchanged into 20 mM HEPES pH 8.0, 150 mM NaCl, and 2 mM DTT for GraFix (Kastner et al, 2008; Stark, 2010). A gradient was formed with buffers A, 20 mM HEPES pH 8.0, 150 mM NaCl, 2 mM DTT, and 5% sucrose and B, 20 mM HEPES pH 8.0, 150 mM NaCl, 2 mM DTT, 25% sucrose, and 0.1% glutaraldehyde using the Gradient Master (BioComp Instruments). In total, 500 μl of the sample was applied on top of the gradient, and the tubes were centrifuged at 40,000 rpm at 4 °C using a Beckman SW40 rotor for 16 h. The gradient was fractionated in 500-μl fractions from top to bottom, and the fractions were analysed by SDS-PAGE with Coomassie blue staining and negative staining EM.

## Negative staining sample preparation, data collection and processing

Copper grids, 300 mesh, with continuous carbon layer (TAAB) were glow-discharged using the PELCO easiGlow™ system (Ted Pella). GraFix fractions with and without dialysis were used. Dialysed fractions were diluted to 0.02 mg/ml. In all, 4 μl of sample were adsorbed for 2 min onto the carbon side of the glow-discharged grids, then the excess was side blotted with filter paper. The grids were washed in two 15-μl drops of buffer and one 15 μl drop of 2% uranyl acetate, blotting the excess between each drop, and then incubated with a 15 μl drop of 2% uranyl acetate for 2 min. The excess was blotted by capillary action using a filter paper, as previously described (Scarff et al, 2018).

The grids were loaded into a Tecnai F20 (Thermo Fisher Scientific) electron microscope, operated at 200 kV, field emission gun (FEG), with pixel size of 1.48 Å. Micrographs were recorded using an 8k × 8k CMOS F816 camera (TVIPS) at a defocus range of −0.8 to −2 μm. For Mis18α/β/Mis18BP1$_{20-130}$ (Mis18$_{core}$), 163 micrographs were recorded and analysed using CryoSPARC 3.1.0 (Punjani et al, 2017). The contrast transfer function (CTF) was estimated using Gctf (Zhang, 2016). Approximately 750 particles were manually picked and submitted to 2D classification. The class averages served as templates for automated particle picking. Several rounds of 2D classification were employed to remove bad particles and assess the data, reducing the 14,840 particles to 5540. These were used to generate three ab initio models followed by homogeneous refinement with the respective particle sets.

## CLMS

Cross-linking was performed on gel-filtered complexes dialysed into PBS. In total, 16 μg EDC and 35.2 μg sulpho-NHS were used to cross-link 10 μg of Mis18α/β with Mis18BP1$_{20-130}$ (Mis18$_{core}$) for 1.5 h at RT. The reactions were quenched with final concentration 100 mM Tris-HCl before separation on Bolt™ 4–12% Bis-Tris Plus gels (Invitrogen). Sulfo-SDA (sulfosuccinimidyl 4,4'-azipentanoate) (Thermo Scientific Pierce) cross-linking reaction was a two-step process. First, sulfo-SDA mixed with Mis18α/β (0.39 μg/μl) at different ratio (w/w) of 1:0.07, 1:0.13, 1:0.19, 1:0.38, 1:0.5, 1:0.75, 1:1 and 1:1.4 (Mis18α/β:Sulfo-SDA) was allowed to incubate 30 min at room temperature to initiate incomplete lysine reaction with the sulfo-NHS ester component of the cross-linker. The diazirine group was then photoactivated for 20 min using UV irradiation from a UVP CL-1000 UV Cross-linker (UVP Inc.) at 365 nm (40 W). The reactions were quenched with 2 μl of 2.7 M ammonium bicarbonate before loading on Bolt™ 4–12% Bis-Tris Plus gels (Invitrogen) for separation. Following previously established protocol (Maiolica et al, 2007), either the whole sample or specific bands were excised, and proteins were digested with 13 ng/μl trypsin (Pierce) overnight at 37 °C after being reduced and alkylated. The digested peptides were loaded onto C18-Stage-tips (Rappsilber et al, 2007) for LC-MS/MS analysis.

LC-MS/MS analysis was performed using an Orbitrap Fusion Lumos (Thermo Fisher Scientific) coupled online with an Ultimate 3000 RSLCnano system (Thermo Fisher Scientific) with a "high/high" acquisition strategy. The peptide separation was carried out on a 50-cm EASY-Spray column (Thermo Fisher Scientific). Mobile phase A consisted of water and 0.1% v/v formic acid. Mobile phase B consisted of 80% v/v acetonitrile and 0.1% v/v formic acid. Peptides were loaded at a flow rate of 0.3 μl/min and eluted at 0.2 μl/min or 0.25 μl/min using a linear gradient going from 2% mobile phase B to 40% mobile phase B over 109 or 79 min, followed by a linear increase from 40% to 95% mobile phase B in 11 min. The eluted peptides were directly introduced into the mass spectrometer. MS data were acquired in the data-dependent mode with a 3 s acquisition cycle. Precursor spectra were recorded in the Orbitrap with a resolution of 120,000. The ions with a precursor charge state between 3+ and 8+ were isolated with a window size of 1.6 $m/z$ and fragmented using high-energy collision dissociation (HCD) with a collision energy of 30. The fragmentation spectra were recorded in the Orbitrap with a resolution of 15,000. Dynamic exclusion was enabled with a single repeat count and 60-s exclusion duration. The mass spectrometric raw files were processed into peak lists using ProteoWizard (version 3.0.20388) (Kessner et al, 2008), and cross-linked peptides were matched to spectra using Xi software (version 1.7.6.3) (Mendes et al, 2019) (https://github.com/Rappsilber-Laboratory/XiSearch) with in-search assignment of monoisotopic peaks (Lenz et al, 2018). Search parameters were MS accuracy, 3 ppm; MS/MS accuracy, 10 ppm; enzyme, trypsin; cross-linker, EDC; max missed cleavages, 4; missing monoisotopic peaks, 2. For EDC search cross-linker, EDC; fixed modification, carbamidomethylation on cysteine; variable modifications, oxidation on methionine. For sulfo-SDA search: fixed modifications, none; variable modifications, carbamidomethylation on cysteine, oxidation on methionine, SDA-loop SDA cross-link within a peptide that is also cross-linked to a separate peptide. Fragments b and y type ions (HCD) or b, c, y, and z type ions (EThcD) with loss of $H_2O$, $NH_3$ and $CH_3SOH$. 5% on link level False discovery rate (FDR) was estimated based on the number of decoy identification using XiFDR (Fischer and Rappsilber, 2017).

## Integrative structure modelling

### Input subunits

Using the Mis18α_Yippee as a template, we generated high-confidence structural models for the Mis18α and Mis18β Yippee domains (using the homology modelling server Phyre2, www.sbg.bio.ic.ac.uk/phyre2/ (Kelley et al, 2015)). These models were almost identical with those obtained using Raptorx (http://raptorx6.uchicago.edu/) and Alpha-Fold2 (Jumper et al, 2021); structure prediction programmes that employ deep learning approach independent of co-evolution information (Källberg et al, 2012) (Fig. 1E).

### Scoring function for CLMS

A cross-link was considered satisfied if the Calpha-Calpha distance was less than 22 Å. The final score was the fraction of satisfied cross-links.

### Sampling

To determine the structure of the Mis18 complex, we used XlinkAssembler, an algorithm for multi-subunit assembly based on combinatorial docking approach (Inbar et al, 2005; Schneidman-Duhovny and Wolfson, 2020). The input to XlinkAssembler is N subunit structures and a list of cross-links. First, all subunit pairs are docked using cross-links as distance restraints (Schneidman-Duhovny et al, 2005). Pairwise docking generates multiple docked configurations for each pair of subunits that satisfy a large fraction of cross-links (>70%). Second, the combinatorial assembler hierarchically enumerates pairwise docking configurations to generate larger assemblies that are consistent with the CLMS data.

XlinkAssembler was used with 11 subunits to generate a model for Mis18α/β: initial hexamer structure based on AlphaFold (Jumper et al, 2021), two Mis18α_Yippee domains as well as four copies of the two helices in the Mis18α N-terminal helical region (residues 37–55 and 60–76). For docking Mis18BP1 helices, XlinkAssembler was used with 4 subunits: the Mis18α/β_Yippee domains heterodimer and the three Mis18BP1 helices predicted by AlphaFold (residues 21–33, 42–50 and 90–111).

## Cell culture and transfection

The cell line HeLa Kyoto, HeLa 3–8 (having an alphoid^tetO array integrated into one of its chromosome arms) (Ohzeki et al, 2012), as well as HeLa CENP-A-SNAP, GFP-Mis18BP1 inducible CENP-A-SNAP, Mis18β-GFP CENP-A-SNAP and mCherry-Mis18α CENP-A-SNAP (kind gift from Iain Cheeseman (McKinley and Cheeseman, 2014)) were maintained in DMEM (Gibco) containing 10% FBS (Biowest) and 1× Penicillin/Streptomycin antibiotic mixture (Gibco). The cells were incubated at 37 °C in a $CO_2$ incubator in humid condition containing 5% $CO_2$. GFP-Mis18BP1 was induced with 10 μg/ml doxycycline for 18 h. siRNAs (AllStars Negative Control siRNA 1027280. Mis18α: ID s28851, Mis18β: ID s22367; Silencer® Select, Thermo Fisher Scientific) were used in the rescue assays by transfecting the cells using jetPRIME® (Polyplus transfection®) reagent according to the manufacturer's instructions. Briefly, HeLa CENP-A-SNAP, Mis18β-GFP CENP-A-SNAP, GFP-Mis18BP1 inducible CENP-A-SNAP and mCherry-Mis18α CENP-A-SNAP cells were seeded in 12-well plates and incubated overnight. siRNAs (50 pmol), vectors (200 ng) and the jetPRIME® reagent were diluted in the jetPRIME® buffer, vortexed and spun

down. The transfection mixture was incubated for 15 min before adding to the cells in a drop-by-drop manner. The cells were then incubated for 48 h.

The TetR-eYFP tagged proteins were transfected using the XtremeGene-9 (Roche) transfection reagent according to the manufacturer's protocol. The HeLa 3–8 cells attached onto the coverslip in a 12-well plate were transfected with the corresponding vectors (500 ng) and the transfection reagent diluted in Opti-MEM (Invitrogen) followed by incubation for 36–48 h.

## Generation of monoclonal antibodies against Mis18α/Mis18β

Lou/c rats and C57BL/6J mice were immunised with 60 μg purified recombinant human Mis18α/β protein complex, 5 nmol CpG (TIB MOLBIOL, Berlin, Germany), and an equal volume of Incomplete Freund's adjuvant (IFA; Sigma, St. Louis, USA). A boost injection without IFA was given 6 weeks later and 3 days before fusion of immune spleen cells with P3X63Ag8.653 myeloma cells using standard procedures. Hybridoma supernatants were screened for specific binding to Mis18α/β protein complex and also for binding to purified GST-Mis18β protein in ELISA assays. Positive supernatants were further validated by western blot analyses on purified recombinant human Mis18α/β complex, on cell lysates from *Drosophila* S2 cells overexpressing human Mis18α and on HEK293 cell lysates. Hybridoma cells from selected supernatants were subcloned at least twice by limiting dilution to obtain stable monoclonal cell lines. Experiments in this work were performed with hybridoma supernatants mouse anti-Mis18α (clone 25G8, mouse IgG2b/к) and rat anti-Mis18β (clone 24C8; rat IgG2a/к).

## Western blot

To study the efficiency of DNA and siRNA transfected, HeLa cells were transfected as stated above. Protein was extracted with RIPA buffer and analysed by SDS-PAGE followed by wet transfer using a Mini Trans-Blot® Cell (BioRad). Antibodies used for Western blots were: mouse Mis18α (25G8), rat Mis18β (24C8) (1:100, Helmholtz Zentrum München), Mis18BP1 (1:500, PA5-46777, Thermo Fisher Scientific or 1 μg/ml, ab89265, Abcam), GFP (1:5000, ab290, Abcam), mCherry (1:1000, ab167453, Abcam) and tubulin (1:2000, T5168, Sigma). Secondary antibodies used were ECL Rabbit IgG, ECL Mouse IgG and ECL Rat IgG (1:5000, NA934, NA931, NA935, GE Healthcare), and immunoblots were imaged using NuGlow ECL (Alpha Diagnostics). For imaging with the Odyssey® CLx system, goat anti-mouse 680 and donkey anti-rabbit 800 secondary antibodies were used (1:5000, 926-68070, 926-32213, LI-COR).

## Co-immunoprecipitation

HeLa Kyoto cells were seeded in 100-mm dishes. The cells were depleted of the endogenous Mis18α or Mis18β by siRNA transfection with jetPRIME® (Polyplus transfection®) and simultaneously rescued with siRNA-resistant versions of WT or mutant Mis18α-mCherry and Mis18β-GFP. The cells were harvested after 48 h and lysed by resuspending in immunoprecipitation buffer, 75 mM HEPES pH 7.5, 1.5 mM EGTA, 1.5 mM $MgCl_2$, 150 mM NaCl, 10% glycerol, 0.1% NP40, 1 mM PMSF, 10 mM NaF, 0.3 mM Na-vanadate and cOmplete™ Mini Protease Inhibitor; adapted from (Pan et al, 2017). Cells were

incubated with mixing for 30 min at 4 °C before sonicating with a Bioruptor® Pico (Diagenode). Lysates were then spun for 10 min at $15,000 \times g$. The protein concentrations were determined and adjusted to the same concentration. Protein was taken for inputs, and the rest was incubated with Protein G Mag Sepharose® (GE Healthcare), previously coupled to Mis18α antibody, for 1 h at 4 °C. Next, the bound fraction was separated from unbound by bind beads to the magnet and washing three times with the IP buffer with either 150 mM or 300 mM NaCl. The protein was extracted from the beads by boiling with SDS-PAGE loading dye for 5 min and were analysed by SDS-PAGE followed by western blotting with anti-mCherry, GFP and tubulin antibodies.

### Immunofluorescence and quantification

The transfected cells were washed with PBS and fixed in 4% paraformaldehyde for 10 min, followed by permeabilisation in PBS with 0.5% Triton™ X-100 (Sigma) for 5 min. The cells were then blocked in 3% BSA containing 0.1% Triton™ X-100 for 1 h at 37 °C. The blocked cells were subsequently stained with the indicated primary antibodies for 1 h at 37 °C followed by secondary antibody staining under similar conditions. The following primary antibodies were used for immunofluorescence: anti-ACA (1:300; 15-235; Antibodies Inc.) and anti-CENP-A (1:100, MA 1-20832, Thermo Fisher Scientific). The secondary antibodies used were Alexa Fluor® 488 AffiniPure donkey anti-human IgG, Cy5-conjugated AffiniPure donkey anti-human, and TRITC-conjugated AffiniPure donkey anti-mouse (1:300; 709-546-149, 709-175-149, 715-025-150, Jackson Immunoresearch). Vector shield with DAPI (Vector Laboratories) was used for DNA staining.

Micrographs were acquired at the Centre Optical Instrumentation Laboratory on a DeltaVision Elite™ system (Applied Precision) or Nikon Ti2 inverted microscope. Z stacks were obtained at a distance of 0.2 μm and were deconvolved using SoftWoRx, or AutoQuant software, respectively, followed by analysis using Image J software. The intensity at the tethering site was obtained using a custom-made plugin. Briefly, the Mis18BP1$_{20-130}$-mCherry signal at the tethering site (eYFP) was found for every z-section within a 7-square pixel box. The mean signal intensity obtained was subtracted from the minimum intensities within the section. The values were obtained from a minimum of three biological repeats. Prism 9.1.2 was used to establish if data was normally distributed, before the statistical significance of the difference between normalised intensities at the centromere and tethering region was established by a Mann–Whitney $U$ two-tailed test.

### SNAP-CENP-A assay and quantification

SNAP-CENP-A quench pulse labelling was done as described previously (Jansen et al, 2007). Briefly, the existing CENP-A was quenched by 10 μM SNAP-Cell® Block BTP (S9106S, New England Biolabs). The cells were treated with 1 μM STLC for 15 h for enriching the mitotic cell population, and the newly formed CENP-A was pulse labelled with 3 μM SNAP-Cell® 647-SiR (S90102S, New England Biolabs), 2 h after release from the STLC block (early G1). After pulse labelling, the cells were washed, fixed and processed for immunofluorescence. Images were obtained using DeltaVision Elite™ system (Applied Precision), deconvolved by SoftwoRx and processed by Image J. The average centromere intensities were obtained using a previously described macro CraQ (Bodor et al,

2012). Briefly, the centromeres were defined by a $7 \times 7$ pixel box using a reference channel, and the corresponding mean signalling intensity at the data channel was obtained by subtracting the minimum intensities within the selection. The values plotted were obtained from a minimum of three independent experiments. After testing is the data was normally distributed, the statistical significance of the difference between normalised intensities at the centromere region was established by a Mann–Whitney $U$ test using Prism 9.1.2.

## Data availability

PDB ID: 7SFY for Mis18α/β$_{C-term}$: https://www.rcsb.org/structure/7SFY. PDB ID: 7SFZ for Mis18α$_{Yippee}$: https://www.rcsb.org/structure/7SFZ. The mass spectrometry proteomics data have been deposited in the ProteomeXchange Consortium via the PRIDE (Perez-Riverol et al, 2019) partner repository: identifier PXD047345. Access codes for the EM density maps deposited in EMDB: EMD-50218, EMD-50219, EMD-50220. Access code for the integrative structure model deposited in PDB-Dev: PDBDEV_00000380. Access code for the original microscopy images deposited in BioImage Archive: S-BIAD1181. Plugin for analysing intensities at tethering site deposited in Zenodo: https://doi.org/10.5281/zenodo.5708337 (https://zenodo.org/records/5708337).

The source data of this paper are collected in the following database record: biostudies:S-SCDT-10_1038-S44319-024-00183-w.

## Peer review information

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

## Acknowledgements

The authors would like to thank David Kelly from the Centre Optical Instrumentation Laboratory, Martin Wear from the Edinburgh Protein Production Facility as well as Marcus Wilson and Maarten Tuijtel of the Cryo-Electron Microscopy Facility for their help. The authors also thank Diamond Light Source and the staff of beamline B21 (proposal sm23510), as well as Advanced Photon Source and the staff at beamlines LS-CAT 21 ID-G and ID-D. Thanks also to Iain Cheeseman for the kind gift of cell lines. The Wellcome Trust generously supported this work through Senior Research Fellowships to AA Jeyaprakash (202811), J Rappsilber (084229), O Davies (219413/Z/19/Z) and P Heun (103897/Z/14/Z), a Centre Core Grant (092076 and 203149) and an instrument grant (108504) to the Wellcome Trust Centre for Cell Biology. AA Jeyaprakash and his team are also funded by the European Union (ERC Advanced Grant, CHROMSEG, 101054950) and the Medical Research Council (MRC, United Kingdom; MR/X0012451/1). Views and opinions expressed are, however, those of the authors only and do not necessarily reflect those of the European Union or the European Research Council. Neither the European Union nor the granting authority can be held responsible for them. The NIH supported the work of U Cho and SH Park (NIH/NIDDK; R01 DK111465). The work of D Schneidman-Duhovny is supported by ISF 1466/18 and the Israeli Ministry of Science and Technology.

## Author contributions

**Reshma Thamkachy**: Formal analysis; Investigation; Writing—review and editing. **Bethan Medina-Pritchard**: Data curation; Formal analysis; Investigation; Writing—original draft; Writing—review and editing. **Sang Ho Park**: Investigation. **Carla G Chiodi**: Formal analysis; Investigation. **Juan Zou**: Formal analysis; Investigation. **Maria de la Torre-Barranco**: Formal analysis; Investigation. **Kazuma Shimanaka**: Investigation. **Maria Alba Abad**: Investigation. **Cristina Gallego Páramo**: Formal analysis. **Regina Feederle**: Resources. **Emilija Ruksenaite**: Investigation. **Patrick Heun**: Resources; Supervision. **Owen R Davies**: Formal analysis. **Juri Rappsilber**: Resources. **Dina Schneidman-Duhovny**: Data curation; Formal analysis; Writing—original draft. **Uhn-Soo Cho**: Conceptualisation; Resources; Supervision. **A Arockia Jeyaprakash**: Conceptualisation; Resources; Supervision; Writing—original draft; Writing—review and editing.

Source data underlying figure panels in this paper may have individual authorship assigned. Where available, figure panel/source data authorship is listed in the following database record: biostudies:S-SCDT-10_1038-S44319-024-00183-w.

## Disclosure and competing interests statement

The authors declare no competing interests.

# Expanded View Figures

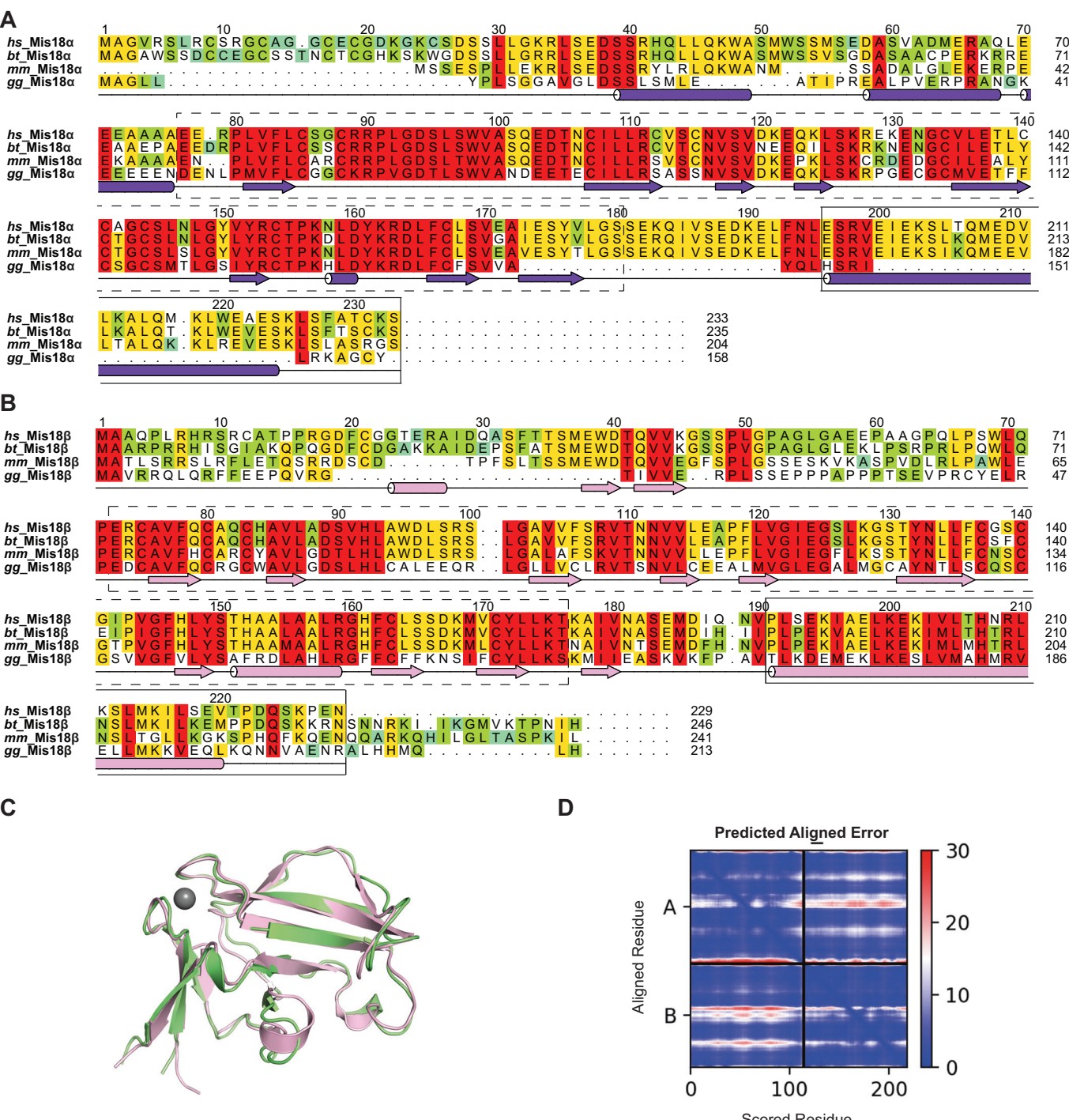

**Figure EV1.   Mis18α and Mis18β contain two domains capable of oligomerising.**

(A, B) Domain architecture and amino acid conservation of (A) Mis18α and (B) Mis18β. Alignments include *Homo sapiens* (*hs*), *Bos taurus* (*bt*), *Mus musculus* (*mm*) and *Gallus gallus* (*gg*). The conservation score is mapped from red to cyan, where red corresponds to highly conserved and cyan to poorly conserved. Secondary structures as annotated/predicted by Conserved Domain Database [CDD] and PsiPred, http://bioinf.cs.ucl.ac.uk/psipred. Multiple sequence alignments were performed with MUSCLE (Madeira et al, 2019) and edited with Aline (Bond and Schüttelkopf, 2009). Dashed boxes highlight Yippee domains whilst solid boxes highlight C-terminus α-helices. (C) Superposition of Mis18β_Yippee structures predicted by AlphaFold (light pink) and RaptorX (green). RaptorX generated five models and the model with the lowest estimated error (1.9 Å) is shown here. The AlphaFold and RaptorX models superpose well with an RMSD of 0.95 Å. (D) The PAE plot corresponding to the Mis18α/β_Yippee AlphaFold model shown in Fig. 1D.

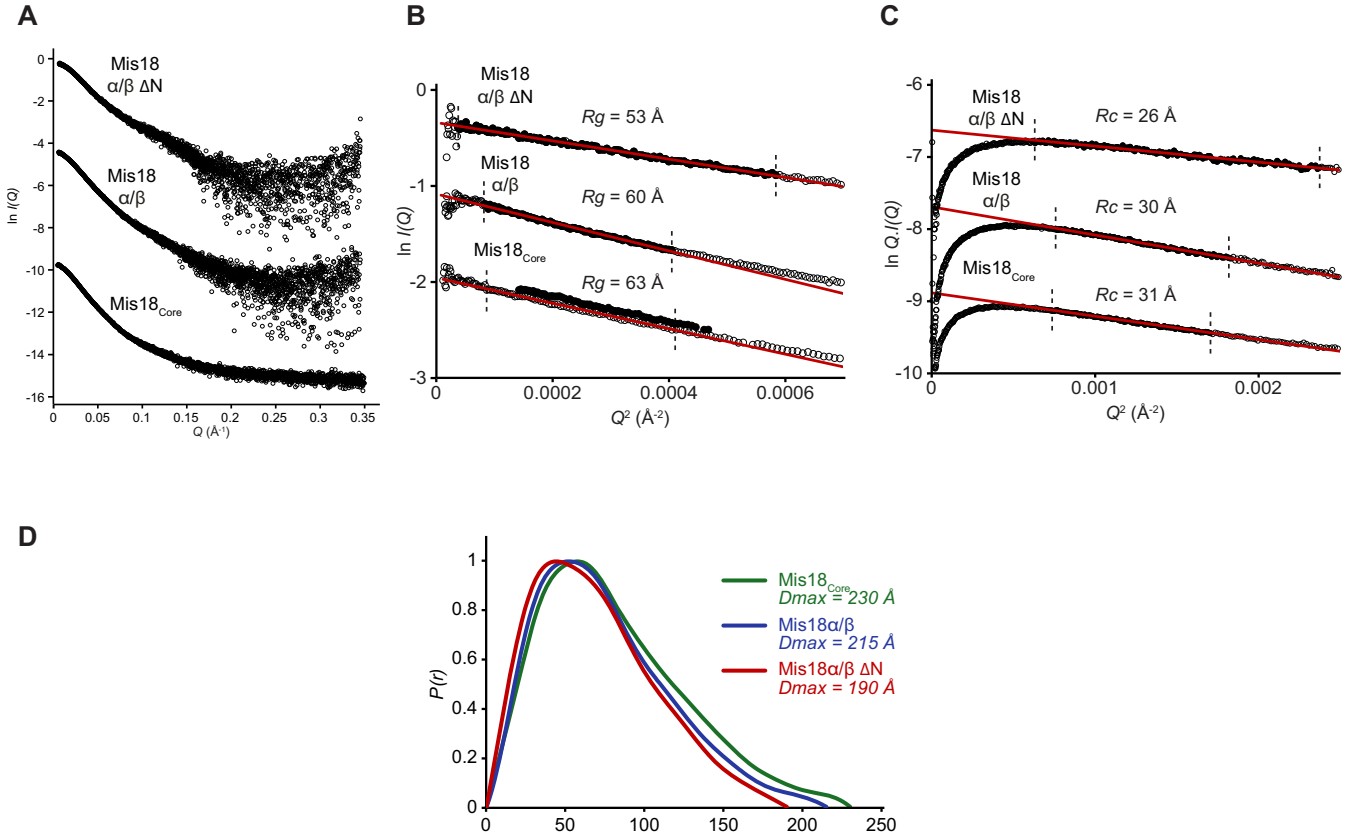

**Figure EV2.  SAXS analysis of Mis18α/β ΔN, Mis18α/β and Mis18core.**

(A) SAXS scattering curves of Mis18α/β ΔN, Mis18α/β and Mis18core. (B) Guinier Plot showing *Rg* of 53 Å, 60 Å, and 63 Å for Mis18α/β ΔN, Mis18α/β and Mis18corer, respectively. (C) Modified Guinier Plot showing *Rc* of 26 Å, 30 Å, and 31 Å for Mis18α/β ΔN, Mis18α/β and Mis18core, respectively. (D) SAXS *P(r)* distributions showing maximum dimensions of 190 Å, 215 Å, and 230 Å for Mis18α/β ΔN, Mis18α/β and Mis18core, respectively.

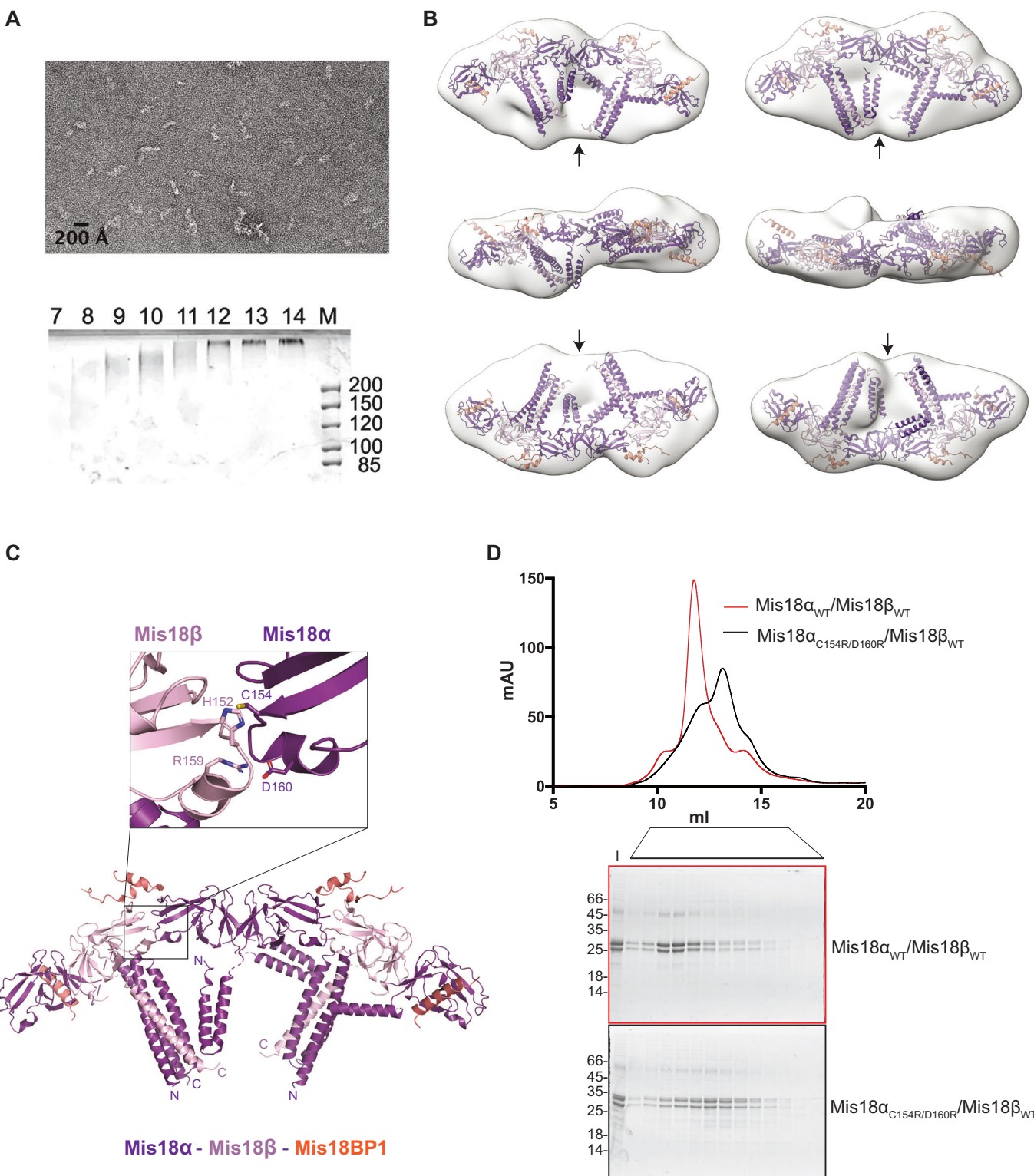

**Figure EV3.  Structural characterisation of the Mis18$_{core}$ complex.**

(A) Representative micrograph of negative staining EM of the Mis18α/Mis18β/Mis18BP1$_{20-130}$ (Mis18$_{core}$) complex cross-linked using GraFix (Kastner et al, 2008; Stark, 2010). Beneath is the corresponding SDS-PAGE analysis of fractions from GraFix, fractions 8 and 9 were used to make grids. (B) Two models (Class II-III) generated for Mis18$_{core}$ from negative staining EM analysis. All show that the overall shapes of the Mis18$_{core}$ resemble a telephone handset with 'ear' and 'mouth' pieces assuming different relative orientations. (C) Cartoon representation of the model of Mis18$_{core}$ complex generated in Fig. 2B. Zoomed in panel shows interaction between Mis18α and Mis18β Yippee domains using the second interface. Important residues for this interaction highlighted in pink and purple. (D) SEC profile of Mis18α$_{WT}$/Mis18β$_{WT}$ (red) and Mis18α$_{C154R/D160R}$/Mis18β$_{WT}$ (black) and corresponding SDS-PAGE analysis of the fractions. Samples were analysed using Superdex 200 increase 10/300 in 20 mM Tris-HCl pH 8.0, 250 mM NaCl and 2 mM DTT.

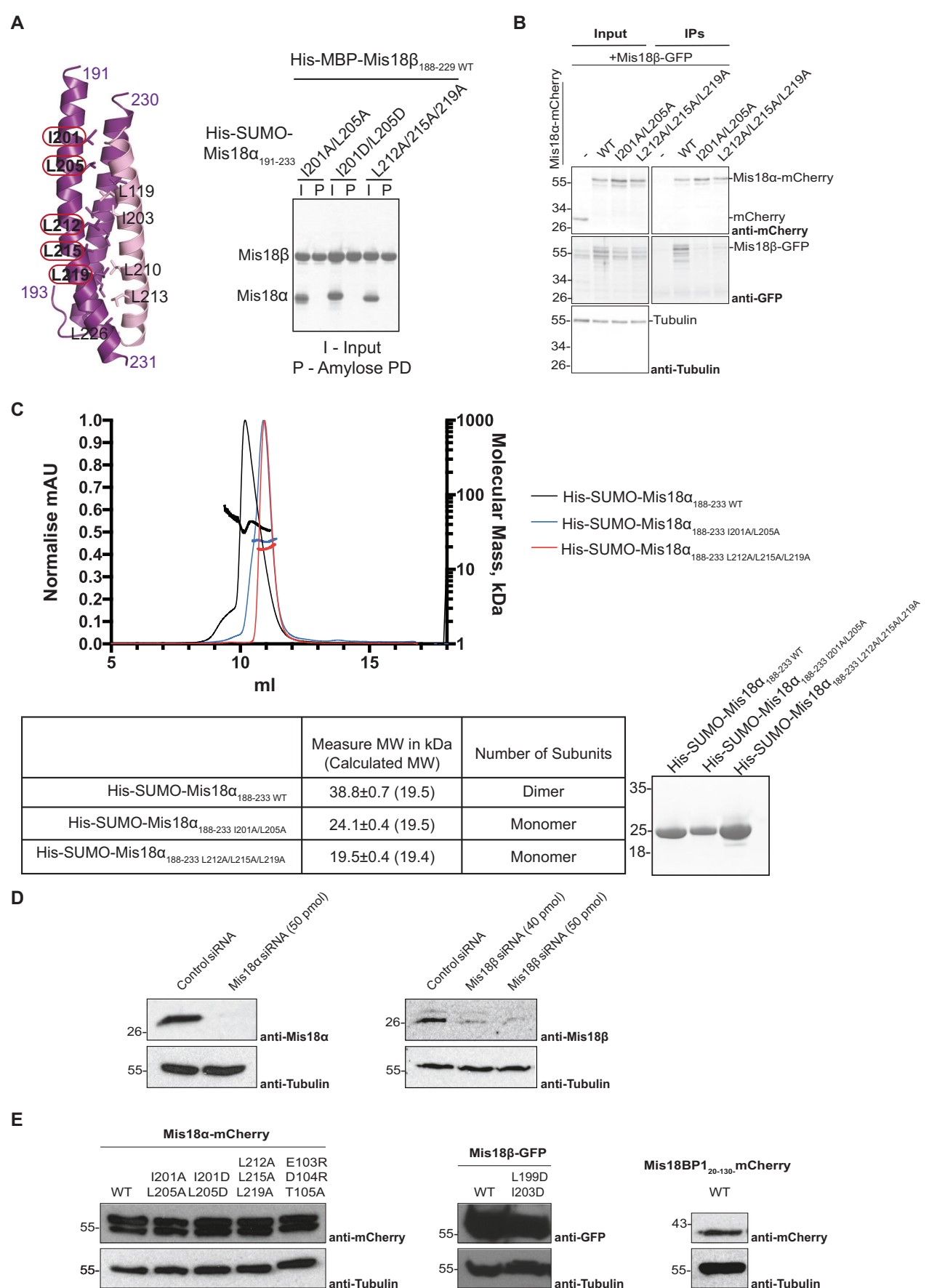

**Figure EV4. Structural and biochemical characterisation of Mis18α C-terminal helix.**

(A) Cartoon representation of the crystal structure of Mis18α$_{C\text{-term}}$/Mis18β$_{C\text{-term}}$ (PDB ID: 7SFY). Mis18α is shown in purple and Mis18β in light pink. Potential residues involved in the interaction are highlighted. Mis18α (purple) and Mis18β (light pink). Right panel shows SDS-PAGE analysis of cobalt and amylose pull-down of His-MBP-Mis18β$_{188\text{-}229\ WT}$ with His-SUMO-Mis18α$_{191\text{-}233}$ mutants. SDS-PAGE shows protein bound to nickel resin as input (I) and protein-bound to amylose resin to assess interaction (P). Control with WT proteins shown in Fig. 4A. (B) Western blot analysis of co-immunoprecipitation (Co-IP) experiments using Mis18α antibody to test interaction of mCherry as a control, Mis18α−mCherry with and without mutations in the C-terminal α-helices and Mis18β-GFP. Top panel shows blot against mCherry, middle panel shows blot against GFP, and bottom panel shows blot against tubulin as loading control. (C) SEC-MALS of His-SUMO-Mis18α$_{188\text{-}233\ WT}$, His-SUMO-Mis18α$_{188\text{-}233\ I201A/L205A}$ and His-SUMO-Mis18α$_{188\text{-}233\ L212A/L215A/L219A}$. Normalised absorption at 280 nm (mAU, left y-axis) and molecular mass (kDa, right y-axis) are plotted against elution volume (ml, x-axis). Measured molecular weight (MW) and the calculated subunit stoichiometry based on the predicted MW. Samples were analysed using a Superdex 75 increase in 50 mM HEPES pH 8.0, 150 mM NaCl and 1 mM TCEP. (D) Representative immunoblots showing expression levels of endogenous proteins after treatment with siRNA. (E) Representative immunoblots showing expression levels of transiently expressed tagged proteins after transfection.

