## [Peer Review File · EMBO Reports]

Structural Basis for Mis18 Complex Assembly and its Implications for Centromere Maintenance

Reshma Thamkachy, Bethan Medina-Pritchard, Sang Ho Park, Carla Chiodi, Juan Zou, Maria de la Torre-Barranco, Kazuma Shimanaka, Maria Abad, Cristina Gallego Páramo, Regina Feederle, Emilija Ruksenaite, Patrick Heun, Owen Davies, Juri Rappsilber, Dina Schneidman-Duhovny, Uhn-soo Cho, and A. Arockia Jeyaparakash

Corresponding author(s): A. Arockia Jeyaparakash (jeyaparakash.arulanandam@ed.ac.uk) , Uhn-soo Cho (uhns00@med.umich.edu)

Review Timeline:

Transfer Date:	22nd Mar 24
Editorial Decision:	24th Apr 24
Revision Received:	6th May 24
Accepted:	6th Jun 24

Editor: Deniz Senyilmaz Tiebe

Transaction Report: This manuscript was transferred to EMBO reports following peer review at Review Commons.

**Review
COMMONS**

Review #1

1. Evidence, reproducibility and clarity:

Evidence, reproducibility and clarity (Required)

Summary

Maintenance of the histone H3 variant CENP-A at centromeres is necessary for proper kinetochore assembly and correct chromosome segregation. The Mis18 complex recruits the CENP-A chaperone HJURP to centromeres to facilitate CENP-A replenishment. Here the authors characterise the Mis18 complex using hybrid structural biology, and determine complex interface separation-of-function mutants.

Major Comments

The SAXS and EM data on the full-length Mis18 components must be included in the main Figures, either as an additional figure or by merging/rearranging the existing figures. The authors discuss these results in three whole paragraphs, which are a very important part of the paper.

Could the authors also compare the theoretical SAXS scattering curves generated by their final model(s) with the experimental SAXS curves? This would provide some additional evidence for the overall shape of their complex model beyond the consistency with the D_{max}/R_g .

Minor Comments

While the introduction is clearly written, an additional cartoon schematic, representing the system/question would be helpful to a non-specialist reader to interpret the context of the study.

No doubt the authors had a reason for choosing their figure allocation, but I wonder if more material couldn't be brought from the supplementaries into the main figures?

Page 6 "Mis18-alpha possesses an additional alpha-helical domain" - please make it clear in addition to what (I assume it's in addition to Mis18-beta).

Page 7 - Report the RMSD of the Pombe vs. Human Mis18-alpha yipee structures?

Page 7 - "We generated high-confidence structural models...." is there a metric for the confidence as reported by RaptorX? Perhaps including the PAE plots in the supplementary for the AlphaFold generated models would be useful?

Figure 1 - Perhaps label figure 1b as being experimentally determined, with the R values (as for Figure 1d), and 1c being a predicted model.

Page 8 "This observation is consistent with the theoretically calculated pI of the Mis18alpha helix" This is a circular argument, of course this region has a low pI due to the amino acid composition. Please remove this statement.

Page 8 "...reveals tight hydrophobic interactions" these are presumably shown in Figure 1d rather than in the referenced 1e.

Page 8 - The authors should briefly somewhere discuss why there is a difference between their results and those in Pan et al 2009. As I understand it, the Pan et al paper was based in part on modelling with CLMS data as restraints.

Figure 1 - The labelling of the residues for Mis18-alpha in Figure 1d is problematic, they are black on dark purple (might be my printer/screen/eyes) suggest amending.

Figure S3a - Do the authors have some data to show the mass of the cross-linked complex that was loaded onto grids is consistent with what is expected?

Figure S3b - scale bar

Figure S3c - Could the authors show or explain the differences between these different 3D reconstructions?

Page 9 - The use of "AFM" for AlphaFoldMultimer" is a little confusing since AFM is the established acronym for Atomic Force Microscopy. Perhaps AF2M?

Figure S4a - Control missing for Mis18-alpha wild-type

Figure S4 d and e - The contrast between the bands and the background is very bad (at least in my copy).

Page 13 "Our structural analysis suggests that two Mis18BP1 fragments.....". How did you arrive at this conclusion? Is this based on the AlphaFold/RaptorX model? What additional evidence do you have that the positioning of the Mis18BP1 is correct? Does the CLMS data support this?

Figure 4a - Would the authors like to consider using a different colour for Mis18BP1? The contrast is not great, especially in the electrostatic surface inset.

2. Significance:

Significance (Required)

General Assessment

The paper is extremely clearly written. Likewise the figures are beautifully presented and the data extremely clean and fully supportive of the authors conclusions. Indeed it is seldom that one sees the depth of the structural approaches (X-ray, CLMS, EM, SAXS) in one paper which is a huge strength of the manuscript. In addition the translation of this data into very clean cell biological experiments, makes the paper truly outstanding.

Advance

The authors provide the first model of the Mis18 complex, with extensive evidence to back up this model. The authors provide additional evidence as to how the deposition/renewal of CENP-A might be mediated by the Mis18 complex. The advance comes from both the level of clarity, detail, and scope achieved in this paper.

Audience

This will likely be of great interest to anyone with an interest in chromosome biology, plus be of interest to structural biologists as an outstanding example of hybrid structural biology.

Expertise

I am a biochemist with a background in structural biology with some familiarity with centromere biology

3. How much time do you estimate the authors will need to complete the suggested revisions:

Estimated time to Complete Revisions (Required)

(Decision Recommendation)

Between 1 and 3 months

Yes

Review #2

1. Evidence, reproducibility and clarity:

Evidence, reproducibility and clarity (Required)

****Summary:****

The manuscript "structural basis for Mis18 complex assembly: implications for centromere maintenance" by Thamkachy and colleagues describes a study that uses structural analysis to test essential candidate residues in Mis18 complex components in CENP-A loading. For chromosomes to faithfully segregate during cell division, CENP-A levels must be maintained at the centromere. How CENP-A levels are maintained is therefore important to understand at the mechanistic level. The Mis18 complex has been found to be important, but how exactly the various Mis18 complex components interact and how they regulate new CENP-A loading remains not fully understood. This study set out to characterize the critical residues using X-ray crystallography, negative staining EM, SEC analysis, molecular modeling (Raptorx, AlphaFold2, and AlphaFold-multimer) to identify the residues of Mis18a and Mis18b that are critical for the formation of the Mis18a/b hetero-hexamers and which residues are important for Mis18a and Mis18BP1 interactions. A complex beta-sheet interface dictates the Mis18a and Mis18b interactions. Mutating the Mis18a residues that are important for the Mis18a/b interactions resulted in impaired pull-down of Mis18b and

reduced centromeric levels of mutated Mis18a. The functional consequences of mutating residues that impair Mis18a/b interactions is that with reduced centromeric levels of Mis18a, also impaired new CENP-A loading. Interestingly, mutated Mis18b did not impact centromeric Mis18a levels and only modestly impaired new CENP-A loading. These data were interpreted that Mis18a is critical for new CENP-A loading, whereas Mis18b might be involved in finetuning how much new CENP-A is loaded. Overall, it is a very well described and well written study with exciting data.

****Major comments:****

- Overall, the structural data and the IF data support the importance of Mis18a residues 103-105 are critical for centromeric localization and new CENP-A loading, whereas Mis18b residues L199 and I203 are critical for centromeric localization, but only very modestly impair centromeric Mis18a localization and new CENP-A loading. In the discussion the authors argue that the N-terminal helical region of Mis18a mediate HJURP binding. This latter is postulated based on published work, but not tested in this work. This should be clarified as such.
- Overall, the authors clearly describe their data and methodology and use adequate statistical analyses. The structural data of the Mis18a/b complex being a hetero-hexamers is convincing, but the validation in vivo is missing. As structural experiments are not performed under physiological conditions, it is important to establish the stoichiometry in vivo to further support the totality of the findings of the structural experiments and modeling. The data for the hierarchical assembly of Mis18a and Mis18b at the centromere and its importance in new CENP-A loading is convincing. An additional open question is whether "old" centromeric CENP-A or HJURP:new CENP-A complex is needed to recruit Mis18a to the centromere and whether the identified residues have a role in Mis18a centromeric localization. These data would provide a solid link between the Mis18 complex and how it is directly linked to new CENP-A loading.

****Minor comments:****

- The bar graphs shown ideally also show the individual data points for the authors to appreciate the spread of the data. These figures can be replicated in the Supplemental to avoid making the main figures look too busy.

2. Significance:

Significance (Required)

- This study uses a broad range of structural techniques, including molecular modeling which were subsequently validated by in vitro pull-down assays, co-IP, and IF. This combination of these techniques is important because many structural techniques cannot be performed under physiological conditions. Validating the main findings of the structural results by IF and co-IP is therefore critical.
- This work greatly advances our structural understanding how Mis18a, Mis18b, and Mis18BP1 form the Mis18 complex and how the critical residues in especially Mis18a help the Mis18 complex localize to the centromere and influence new CENP-A loading. This study also provides the first strong evidence in hierarchical assembly of the Mis18 complex.
- How centromere identity is maintained is a critical question in chromosome biology and genome integrity. The Mis18 complex has been identified as an important complex in the process. Several structural and mutational studies (all adequately cited in this manuscript) have tried to address which residues guide the assembly and functional regions of the Mis18 complex. This work builds and expands our understanding how especially Mis18a holds a pivotal role in both Mis18 complex formation and its impact on maintaining centromeric CENP-A levels.
- This work will be of interest to the chromosome field in general and anyone studying the mechanism of cell division.
- Chromatin, centromere, CENP-A, cell division. This reviewer has limited expertise in structural biology.

3. How much time do you estimate the authors will need to complete the suggested revisions:

Estimated time to Complete Revisions (Required)

(Decision Recommendation)

Between 1 and 3 months

Yes

Review #3

1. Evidence, reproducibility and clarity:

Evidence, reproducibility and clarity (Required)

Centromere identity is defined by CENP-A loading to specific sites on genomic DNA. CENP-A loading is known to rely on the Mis18 complex, and several regulators are known; yet how the Mis18 complex achieves this complex process has remained puzzle. By elucidating the structural basis of Mis18 complex assembly using integrative structural approaches the authors show that multiple homo and heterodimeric interfaces of Mis18 α , beta and Mis18BP1 are involved in centromere maintenance. The authors show that Mis18 α can associate with centromeres and deposit CENP-A independent of Mis18 β . Mis18 α functions in CENP-A deposition at centromeres independent of Mis18 β . Mis18 β is required for maintaining a specific level of CENP-A occupancy at centromeres. Thus, using structure-guided and separation-of-function mutants the study reveals how Mis18 complex ensures centromere maintenance.

****Major comments:****

This is an excellent study on centromere inheritance, combining structural and cell biology techniques. The comments here primarily refer to Cell biology aspect of the work.

1. Figures show that new CENP-A deposits in Mis18 β L199D/I203D mutants, but the level was reduced moderately. Based on this observation, the authors make a strong conclusion that Mis18 β licenses the optimal levels of CENP-A at centromeres. Mis18 α may be essential for both CENP-A incorporation and depositing a specific amount of CENP-A, as Mis18 α and CENP-A levels are both reduced in Mis18 β L199D/I203D mutants which failed to form the triple helical assembly with Mis18 α as shown in Figure 3B and 3C. The authors may want to qualify some of these claims as preliminary or speculative.
2. This work and others show that phosphorylation of Mis18BP1 by CDK1 can interfere with complex function (Spiller et al., 2017, Pan et al., 2017). Does the structure provide any insight into PLK1-mediated phosphorylation surfaces for

activation of the complex? If yes, a brief discussion would help to link CDK1 and PLK1 mediated opposing actions will strengthen the work.

3. I am happy with the way cell biology data and the methods are presented so that they can be reproduced. The experiments are adequately replicated and the statistical analysis adequate. It will help to include sample size of cells or centromeres used for building the graphs.

4. This is a strong interdisciplinary study using a variety of in vitro and in vivo techniques. Can the authors discuss if they expect chromatin associated Mis18 complex to host a similar structure as the soluble one? In other words, are they able to comment on any key differences between chromatin and non-chromatin associated Mis18 complexes.

****Minor comments:****

In cell biology experiments, fluorescence intensities could be presented as a superplot for added value across cells and repeats (instead of bar graphs). More on superplot: <https://doi.org/10.1083/jcb.202001064>.

In general, ACA levels do not appear to change significantly between WT and mutant expressing cells although new CENP-A loading is significantly absent in the presence of a few mutants - please comment if ACA used here can recognise CENP-A. Would this mean that old CENP-A remains normally?

It is unclear whether any of the mutant acted in a dominant negative fashion in the presence of endogenous Mis18 proteins. It would have been useful to test this particularly in the context of mis18alpha mutants that seem to fully abolish new CENP-A recruitment.

In figure 3a, GFP panel (input lane, 1) is shown to mark a band corresponding to GFP. Is this expected? Please comment.

Would be useful to have the scale for the cropped images presented as insets.

Figure 4B should read YFP and not YPF.

The authors may want to explain whether the tag differences matter for their study (Case in point: His-SUMO-Mis18a191-233 WT and mutant His-MBP-Mis18b188-229 proteins).

2. Significance:

Significance (Required)

This work elucidates the structural basis of Mis18 complex assembly and the intermolecular interfaces essential for Mis18 functions. This is a significant advance in the field as it helps researchers in the field better understand CENP-A deposition and mechanism underpinning the maintenance of centromere identity. This is a broad area of research benefitting those studying cell division, genome stability, centromere identity and epigenetics might all be interested in and influenced by these findings. Novelty and strength lies in combining structural and cell biology work.

Strengths of the work are structural details of the Mis18 complex. Minor weakness is the link between Mis18 structure and Centromere inheritance is limited to one immunostaining assay (I have mentioned this as a minor comment because addressing this may not be within the scope of this manuscript and is likely to require a repeat of a vast majority of the work with additional reagents which may not directly add value to the current manuscript).

3. How much time do you estimate the authors will need to complete the suggested revisions:

Estimated time to Complete Revisions (Required)

(Decision Recommendation)

Between 1 and 3 months

No

Full Revision

Manuscript number: RC-2023-02275

Corresponding author(s): A. Arockia, Jeyaprakash

[Please use this template only if the submitted manuscript should be considered by the affiliate journal as a full revision in response to the points raised by the reviewers.

*If you wish to submit a preliminary revision with a revision plan, please use our "Revision Plan" template. **It is important to use the appropriate template to clearly inform the editors of your intentions.**]*

1. General Statements [optional]

This section is optional. Insert here any general statements you wish to make about the goal of the study or about the reviews.

We are thankful to the reviewers for their positive evaluation of our manuscript and constructive suggestions. As discussed below, we have now incorporated their suggestions by performing additional analyses (new figure panels (Fig. 2f and Fig. S1c, Fig. S1d and Fig. S3a)), reorganising figures/adding superplots (Fig. 1a, Fig. 2, Fig. S3, Fig. 5a, Fig. 3, Fig. 4 and Fig. 5), and making textual changes. We believe these modifications adequately address reviewers' suggestions and significantly improve the overall quality of the manuscript.

This section is mandatory. Please insert a point-by-point reply describing the revisions that were already carried out and included in the transferred manuscript.

Reviewer #1 (Evidence, reproducibility and clarity (Required)):

Summary

Maintenance of the histone H3 variant CENP-A at centromeres is necessary for proper kinetochore assembly and correct chromosome segregation. The Mis18 complex recruits the CENP-A chaperone HJURP to centromeres to facilitate CENP-A replenishment. Here the authors characterise the Mis18 complex using hybrid structural biology, and determine complex interface separation-of-function mutants.

Major Comments

The SAXS and EM data on the full-length Mis18 components must be included in the main Figures, either as an additional figure or by merging/rearranging the existing figures. The authors discuss these results in three whole paragraphs, which are a very important part of the paper.

Full Revision

We thank the reviewer for this constructive suggestion. We have now included an additional figure (new Fig. 2, attached below), that highlights the fit of the integrative model against the SAXS and EM data.

Fig. 2

Could the authors also compare the theoretical SAXS scattering curves generated by their final model(s) with the experimental SAXS curves? This would provide some additional evidence for the overall shape of their complex model beyond the consistency with the D_{max}/R_g .

We acknowledge the importance of this suggestion. We have now compared the theoretical SAXS scattering curve of the Mis18 α/β core complex (named Mis18 α/β ΔN), which lacks the flexible elements (disordered regions and the helical region flexibility connected to the Yippee domains). The theoretically calculated SAXS scattering curve of the model matches nicely with the experimental data with χ^2 value of 1.36. This data is now included in new Fig. 2 (Fig. 2f) and is referenced on page 9 line 21.

Minor Comments

While the introduction is clearly written, an additional cartoon schematic, representing the system/question would be helpful to a non-specialist reader to interpret the context of the study.

We have now included a cartoon in the revised Fig. 1 to support the introduction on centromere maintenance and the central role of the Mis18 α/β /BP1 complex in this process. Please find the new Fig. 1 below.

Fig. 1

a

No doubt the authors had a reason for choosing their figure allocation, but I wonder if more material couldn't be brought from the supplementary into the main figures?

As addressed in our response to one of the major comments, we have now moved key CLMS, SAXS and EM data from the supplemental figure into the main figure, new Fig. 2.

Page 6 "Mis18-alpha possesses an additional alpha-helical domain" - please make it clear in addition to what (I assume it's in addition to Mis18-beta).

Apologies for the lack of clarity. We have now rephrased this sentence to highlight that this difference is in comparison with Mis18 β on page 6 line 15.

Page 7 - Report the RMSD of the Pombe vs. Human Mis18-alpha yipee structures?

The *S. pombe* Mis18 Yippee structure superposes on to the Human Mis18 α Yippee domain with an RMSD of 0.92 angstroms with is now mentioned on page 7 line 9.

Page 7 - "We generated high-confidence structural models...." is there a metric for the confidence as reported by RaptorX? Perhaps including the PAE plots in the supplementary for the AlphaFold generated models would be useful?

We thank the reviewer for the valid suggestion. We have now included the PAE plot corresponding to the AlphaFold model in the supplementary Fig. S1d and reference on page 7 line 18. RaptorX ranks models based on estimated error. We have now included this information in the new figure legend for Supplementary Fig. S1.

Figure 1 - Perhaps label figure 1b as being experimentally determined, with the R values (as for Figure 1d), and 1c being a predicted model.

We have included Rfree and Rwork values for the Mis18 α Yippee homo dimer structure and labelled Mis18 α/β Yippee hetero-dimer as the predicted model in Fig. 1c and 1d.

Page 8 "This observation is consistent with the theoretically calculated pl of the Mis18alpha helix" This is a circular argument, of course this region has a low pl due to the amino acid composition. Please remove this statement.

We have now removed this statement as suggested.

Page 8 "...reveals tight hydrophobic interactions" these are presumably shown in Figure 1d rather than in the referenced 1e.

We apologise for the oversight. We have now referred to the correct figure (Fig. 1f in the revised Fig. 1).

Page 8 - The authors should briefly somewhere discuss why there is a difference between their results and those in Pan et al 2009. As I understand it, the Pan et al paper was based in part on modelling with CLMS data as restraints.

We thank the reviewer for this suggestion. According to Pan et al., 2009, the model shown by them was generated using CCBUILDER, and their CLMS data could not differentiate the two models with the 2nd Mis18 α C-terminal helix in either parallel or anti-parallel orientation. We now briefly discuss this on page 8 and line 22 as follows: "Although the Pan et al., 2019 model presented the 2nd Mis18 α in a parallel orientation, they did not rule out the possibility of this

assembling in an anti-parallel orientation within the Mis18 α/β C-terminal helical assembly (Pan et al., 2019).”

Figure 1 - The labelling of the residues for Mis18-alpha in Figure 1d is problematic, they are black on dark purple (might be my printer/screen/eyes) suggest amending.

We have now rearranged the label positions to overcome this issue. For clarity, the labels that could not be moved appropriately are shown in white.

Figure S3a - Do the authors have some data to show the mass of the cross-linked complex that was loaded onto grids is consistent with what is expected?

Unfortunately, the amount of material that we recover after performing GraFix is not sufficient enough to determine the molecular weight of the crosslinked sample by techniques such as SEC-MALS. However, GraFix fractions were analysed by SDS PAGE, and fractions that ran around the expected molecular weight were selected for EM analysis. We have now included the corresponding SDS-PAGE showing the migration of the crosslinked sample analysed by EM (Supplementary Fig. S3a).

Figure S3b - scale bar

Revised Fig. 2d now includes the scale bar shown.

Figure S3c - Could the authors show or explain the differences between these different 3D reconstructions?

The models mainly differ in the relative orientations of the bulkier structural features that are referred to as ‘ear’ and ‘mouth’ pieces of a telephone handset. This has been mentioned in the text, but we note that the figure is not referenced right next to this statement. We have now amended this (Page 9 line 19), and to make it clear, we have also highlighted the difference using an arrowhead in Fig. 2e and S3b. The different orientations are also stated in the corresponding figure legends.

Page 9 - The use of "AFM" for AlphaFoldMultimer" is a little confusing since AFM is the established acronym for Atomic Force Microscopy. Perhaps AF2M?

We have now replaced AFM with AF2M on page 9 to avoid confusion.

Figure S4a - Control missing for Mis18-alpha wild-type

Apology for the confusion, this control is present in Fig. 4a. We have now stated this in the figure legend of S4a for clarity.

Figure S4 d and e - The contrast between the bands and the background is very bad (at least in my copy).

We have now adjusted the contrast of the blots in Fig. S4d and S4e response to this comment.

Page 13 "Our structural analysis suggests that two Mis18BP1 fragments.....". How did you arrive at this conclusion? Is this based on the AlphaFold/RaptorX model? What additional evidence do you have that the positioning of the Mis18BP1 is correct? Does the CLMS data support this?

We confirm that this statement is based on AlphaFold model. We have now explicitly highlighted this on page 14, line 5. As noted in the same paragraph (page 14, line 19), this model agrees with the contacts suggested by the cross-linking mass spectrometry data presented here.

Figure 4a - Would the authors like to consider using a different colour for Mis18BP1? The contrast is not great, especially in the electrostatic surface inset.

In response to this suggestion, the Mis18BP1 helix is now shown in grey in the inset of Fig. 5a.

Reviewer #1 (Significance (Required)):

General Assessment

The paper is extremely clearly written. Likewise the figures are beautifully presented and the data extremely clean and fully supportive of the authors conclusions. Indeed it is seldom that one sees the depth of the structural approaches (X-ray, CLMS, EM, SAXS) in one paper which is a huge strength of the manuscript. In addition the translation of this data into very clean cell biological experiments, makes the paper truly outstanding.

Advance

The authors provide the first model of the Mis18 complex, with extensive evidence to back up this model. The authors provide additional evidence as to how the deposition/renewal of CENP-A might be mediated by the Mis18 complex. The advance comes from both the level of clarity, detail, and scope achieved in this paper.

Audience

This will likely be of great interest to anyone with an interest in chromosome biology, plus be of interest to structural biologists as an outstanding example of hybrid structural biology.

Expertise

I am a biochemist with a background in structural biology with some familiarity with centromere biology

Reviewer #2 (Evidence, reproducibility and clarity (Required)):

Summary:

The manuscript "structural basis for Mis18 complex assembly: implications for centromere maintenance" by Thamkachy and colleagues describes a study that uses structural analysis to test essential candidate residues in Mis18 complex components in CENP-A loading. For chromosomes to faithfully segregate during cell division, CENP-A levels must be maintained at the centromere. How CENP-A levels are maintained is therefore important to understand at the mechanistic level. The Mis18 complex has been found to be important, but how exactly the various Mis18 complex components interact and how they regulate new CENP-A loading remains not fully understood. This study set out to characterize the critical residues using X-ray crystallography, negative staining EM, SEC analysis, molecular modeling (Raptorx, AlphaFold2, and AlphaFold-multimer) to identify the residues of Mis18a and Mis18b that are critical for the formation of the Mis18a/b hetero-hexamers and which residues are important for Mis18a and Mis18BP1 interactions. A complex beta-sheet interface dictates the Mis18a and Mis18b interactions. Mutating the Mis18a residues that are important for the Mis18a/b interactions resulted in impaired pull-down of Mis18b and reduced centromeric levels of mutated Mis18a. The functional consequences of mutating residues that impair Mis18a/b interactions is that with reduced centromeric levels of Mis18a, also impaired new CENP-A loading. Interestingly, mutated Mis18b did not impact centromeric Mis18a levels and only modestly impaired new CENP-A loading. These data were interpreted that Mis18a is critical for new CENP-A loading, whereas Mis18b might be involved in finetuning how much new CENP-A is loaded. Overall, it is a very well described and well written study with exciting data.

Major comments:

- Overall, the structural data and the IF data support the importance of Mis18a residues 103-105 are critical for centromeric localization and new CENP-A loading, whereas Mis18b residues L199 and I203 are critical for centromeric localization, but only very modestly impair centromeric Mis18a localization and new CENP-A loading. In the discussion the authors argue that the N-terminal helical region of Mis18a mediate HJURP binding. This latter is postulated based on published work, but not tested in this work. This should be clarified as such.

We thank the reviewer for this comment. Our very recent study aimed at understanding the licensing role of Plk1, independent of the work reported here, serendipitously has now validated this suggestion and demonstrates that a Plk1-mediated phosphorylation cascade activates the Mis18 α/β complex via a conformational switch of the N-terminal helical region of Mis18 α , which facilitates a robust HJURP-Mis18 α/β interaction (Parashara et al. bioRxiv 2024). An independent study from the Musacchio lab (Conti et al. bioRxiv, 2024) also reports similar findings, mutually strengthening our independent conclusions. Overall, these studies highlight the importance of the critical structural insights into the Mis18 complex this study reports. We now explicitly discuss the validation of our original hypothesis by citing our recent work along with that of the Musacchio lab. The corresponding section of the last paragraph now reads as

follows (page 17 line 10): “Previously published work identified amino acid sequence similarity between the N-terminal region of Mis18 α and R1 and R2 repeats of the HJURP that mediates Mis18 α/β interaction (Pan et al., 2019). Deletion of the Mis18 α N-terminal region enhanced HJURP interaction with the Mis18 complex (Pan et al., 2019). Here, we show that the N-terminal helical region of Mis18 α makes extensive contact with the C-terminal helices of Mis18 α and Mis18 β , which had previously been shown to mediate HJURP binding by Pan et al., 2019. Collectively these observations suggest that the N-terminal region of Mis18 α might directly interfere with HJURP - Mis18 complex interaction. Two independent recent studies (Parashara et al., 2024, Conti et al., 2024) reveal that this is indeed the case and a Plk1-mediated phosphorylation cascade involving several phosphorylation and binding events of the Mis18 complex subunits relieve the intramolecular interactions between the Mis18 α N-terminal helical region and the HJURP binding surface of the Mis18 α/β C-terminal helical bundle. This facilitates robust HJURP-Mis18 α/β interaction *in vitro* and efficient HJURP centromere recruitment and CENP-A loading in cells. Overall, these studies also highlight the importance of the critical structural insights into the Mis18 complex we report here.”

- Overall, the authors clearly describe their data and methodology and use adequate statistical analyses. The structural data of the Mis18a/b complex being a hetero-hexamer is convincing, but the validation *in vivo* is missing. As structural experiments are not performed under physiological conditions, it is important to establish the stoichiometry *in vivo* to further support the totality of the findings of the structural experiments and modeling. The data for the hierarchical assembly of Mis18a and Mis18b at the centromere and its importance in new CENP-A loading is convincing. An additional open question is whether "old" centromeric CENP-A or HJURP:new CENP-A complex is needed to recruit Mis18a to the centromere and whether the identified residues have a role in Mis18a centromeric localization. These data would provide a solid link between the Mis18 complex and how it is directly linked to new CENP-A loading.

We agree that establishing the stoichiometry of Mis18 subunits of the Mis18 complex *in vivo* would be insightful. However, considering that the Mis18 complex assembles in a specific window of the cell cycle (late Mitosis and early G1), we think characterising the stoichiometry in cells is extremely difficult and technically challenging. However, consistent with our structural model, several lines of independent evidence (Pan et al., 2017 and Spiller et al., 2017) using different biophysical methods (Analytical Ultra Centrifugation (Pan et al., 2017), SEC-MALS (Spiller et al., 2017)) showed that recombinantly purified Mis18 complex (irrespective of the expression host, from both *E. Coli* or insect cells) is a hetero-octamer made of a hetero-hexameric Mis18 α/β (4 Mis18 α and 2 Mis18 β) complex bound to two copies of Mis18BP1. These observations suggested that hetero-hexamericisation of the Mis18 α/β complex may be needed to bind and dimerise Mis18BP1 in cells. Previously published cellular studies support the *in vivo* requirement of the hetero-octameric Mis18 assembly as: (i) Perturbing the hetero-hexamericisation of the Mis18 α/β complex (by introducing mutations at the Mis18 α/β Yippee dimerisation interface, which while did not disrupt Mis18 α/β complex formation, perturbed its hetero-hexamericisation and resulted in a hetero-trimeric Mis18 α/β complex made of 2 Mis18 α

and 1 Mis18 β) abolished Mis18BP1 binding *in vitro* and in cells, consequently abolished CENP-A deposition (Spiller et al., 2017) and (ii) artificial dimerisation of Mis18BP1, by expressing Mis18BP1 as a GST-tagged protein, enhanced the centromere localisation of Mis18BP1 highlighting the requirement of Mis18 α/β hexameric assembly mediated dimerization of Mis18BP1 in cells (Pan et al., 2017). While these studies highlighted the importance of maintaining the right stoichiometry (hetero-octamer of 4 Mis18 α , 2 Mis18 β and 2 Mis18BP1), lack of structural information on how this essential biological assembly is established remained a major knowledge gap. Our work presented here fills this critical knowledge gap by showing that a segment of Mis18BP1 (aa 20-51) also binds at the Yippee dimerisation interface. To highlight this, we have included the following statements in the introduction on page 5 and 20 “Perturbing the Yippee domain-mediated hexameric assembly of Mis18 α/β (that resulted in a Mis18 α/β hetero-trimer, 2 Mis18 α and 1 Mis18 β) abolished its ability to bind Mis18BP1 *in vitro* and in cells (Spiller et al., 2017), emphasising the requirement of maintaining correct stoichiometry of Mis18 α/β subunits. Consistent with this, artificial dimerisation of Mis18BP1, by expressing Mis18BP1 as a GST-tagged protein, enhanced the centromere localisation of Mis18BP1 (Pan et al., 2017).” and in the Results section on page 14 line 12: “Mis18BP1₂₀₋₅₁ contains two short β strands that interact at Mis18 α/β Yippee interface extending the six-stranded- β sheets of both Mis18 α and Mis18 β Yippee domains. This provides the structural rationale for why Yippee domains-mediated Mis18 α/β hetero-hexamersation is crucial for Mis18BP1 binding (Spiller et al., 2017).”

Regarding the question “*whether ‘old’ centromeric CENP-A or HJURP:new CENP-A complex is needed to recruit Mis18a centromere localisation and whether identified residues have a role in Mis18a centromere localisation*”: According to the published literature, the Mis18 complex associates with centromeres through interaction with CCAN components CENP-C and CENP-I (Shono et al., 2015, Dambacher et al., 2012, Moree et al., 2011, Hoffmann et al., 2020). Considering CCAN assembles on CENP-A nucleosomes, and HJURP:new CENP-A centromere recruitment depends on the Mis18 complex, it will be reasonable to argue that the ‘old’ centromeric CENP-A contributes to the centromere localisation of the Mis18 complex. Amongst the components of the Mis18 complex, Mis18BP1 and Mis18 β have previously been suggested to interact with CENP-C. Within the Mis18 complex, we (Spiller et al., 2017) and others (Pan et al., 2017) have shown that Mis18 α can directly interact with Mis18BP1, but it does so more efficiently when Mis18 α hetero-oligomerises with Mis18 β via their Yippee domains. Here, our structural analysis mapped the interaction interfaces and showed that Mis18 α residues E103, D104 and T105 contribute to Mis18BP1 binding, as mutating these residues abolishes centromere localisation of Mis18 α (Fig. 5c and 5d). To accentuate our findings, we have now included the following paragraph in the discussion section (page 17 line 26): “One of the key outstanding questions in the field is how does the Mis18 complex associate with the centromere. Previous studies identified CCAN subunits CENP-C and CENP-I as major players mediating the centromere localisation of the Mis18 complex mainly via Mis18BP1 (Shono et al., 2015, Dambacher et al., 2012, Moree et al., 2011), although Mis18 β subunit has also been suggested to interact with CENP-C (Stellfox et al., 2016). Within the Mis18 complex, we and

others have shown that the Mis18 α/β Yippee hetero-dimers can directly interact with Mis18BP1. Here our structural analysis allowed us to map the interaction interface mediating Mis18 α/β -Mis18BP1 binding. Perturbing this interface on Mis18 α completely abolished Mis18 α centromere localisation and reduced Mis18BP1 centromere levels. These observations show that Mis18 α associates with the centromere mainly via Mis18BP1, and assembly of the Mis18 complex itself is crucial for its efficient centromere association, as previously suggested. Future work aimed at characterising the intermolecular contact points between the subunits of the Mis18 complex, centromeric chromatin and CCAN components and understanding if the Mis18 complex undergoes any conformational and/or compositional variations upon centromere association and/or during CENP-A deposition process, will be crucial to delineate the mechanisms underpinning the centromere maintenance.”

Minor comments:

- The bar graphs shown ideally also show the individual data points for the authors to appreciate the spread of the data. These figures can be replicated in the Supplemental to avoid making the main figures look too busy.

We thank the reviewers for this suggestion. Reviewer #3 made a similar comment and suggested we use Superplot, which allows visualisation of individual data points of independent experiments. We have now revised all bar graphs using Superplot to address both reviewers' suggestions.

Reviewer #2 (Significance (Required)):

- This study uses a broad range of structural techniques, including molecular modeling which were subsequently validated by in vitro pull-down assays, co-IP, and IF. This combination of these techniques is important because many structural techniques cannot be performed under physiological conditions. Validating the main findings of the structural results by IF and co-IP is therefore critical.
- This work greatly advances our structural understanding how Mis18a, Mis18b, and Mis18BP1 form the Mis18 complex and how the critical residues in especially Mis18a help the Mis18 complex localize to the centromere and influence new CENP-A loading. This study also provides the first strong evidence in hierarchical assembly of the Mis18 complex.
- How centromere identity is maintained is a critical question in chromosome biology and genome integrity. The Mis18 complex has been identified as an important complex in the process. Several structural and mutational studies (all adequately cited in this manuscript) have tried to address which residues guide the assembly and functional regions of the Mis18 complex. This work builds and expands our understanding how especially Mis18a holds a pivotal role in both Mis18 complex formation and its impact on maintaining centromeric CENP-A levels.
- This work will be of interest to the chromosome field in general and anyone studying the mechanism of cell division.
- Chromatin, centromere, CENP-A, cell division. This reviewer has limited expertise in structural

biology.

Reviewer #3 (Evidence, reproducibility and clarity (Required)):

Centromere identity is defined by CENP-A loading to specific sites on genomic DNA. CENP-A loading is known to rely on the Mis18 complex, and several regulators are known; yet how the Mis18 complex achieves this complex process has remained puzzle. By elucidating the structural basis of Mis18 complex assembly using integrative structural approaches the authors show that multiple homo and heterodimeric interfaces of Mis18 α , beta and Mis18BP1 are involved in centromere maintenance. The authors show that Mis18 α can associate with centromeres and deposit CENP-A independent of Mis18 β . Mis18 α functions in CENP-A deposition at centromeres independent of Mis18 β . Mis18 β is required for maintaining a specific level of CENP-A occupancy at centromeres. Thus, using structure-guided and separation-of-function mutants the study reveals how Mis18 complex ensures centromere maintenance.

Major comments:

This is an excellent study on centromere inheritance, combining structural and cell biology techniques. The comments here primarily refer to Cell biology aspect of the work.

1. Figures show that new CENP-A deposits in Mis18 β L199D/I203D mutants, but the level was reduced moderately. Based on this observation, the authors make a strong conclusion that Mis18 β licenses the optimal levels of CENP-A at centromeres. Mis18 α may be essential for both CENP-A incorporation and depositing a specific amount of CENP-A, as Mis18 α and CENP-A levels are both reduced in Mis18 β L199D/I203D mutants which failed to form the triple helical assembly with Mis18 α as shown in Figure 3B and 3C. The authors may want to qualify some of these claims as preliminary or speculative.

We thank the reviewer for this suggestion. We agree that although the reduction in CENP-A levels upon replacing WT Mis18 β with Mis18 β L199D/I203D is more prominent than the reduction in centromere localised Mis18 α , one cannot completely rule out the contribution of reduced Mis18 α on CENP-A loading. This also raises an interesting possibility where Mis18 β ensures the correct amount of CENP-A deposition by facilitating the optimal level of Mis18 α at centromeres. We now explicitly discuss this in the discussion as follows (page 16 line 26): “Whilst proteins involved in CENP-A loading have been well established, the mechanism by which the correct levels of CENP-A are controlled is yet to be thoroughly explored and characterised. The data presented here suggest that Mis18 β mainly contributes to the quantitative control of centromere maintenance – by ensuring the right amounts of CENP-A deposition at centromeres – and maybe one of several proteins that control CENP-A levels. We also note that the Mis18 β mutant, which cannot interact with Mis18 α , moderately reduced Mis18 α levels at centromeres, and hence, it is possible that Mis18 β ensures the correct level of CENP-A deposition by facilitating optimal Mis18 α centromere recruitment. Future studies will focus on dissecting the mechanisms underlying the Mis18 β -mediated control of CENP-A loading amounts along with any other mechanisms involved.”

2. This work and others show that phosphorylation of Mis18BP1 by CDK1 can interfere with complex function (Spiller et al., 2017, Pan et al., 2017). Does the structure provide any insight into PLK1-mediated phosphorylation surfaces for activation of the complex? If yes, a brief discussion would help to link CDK1 and PLK1 mediated opposing actions will strengthen the work.

As described in our response to the first major comment of Reviewer 2, our very recent study aimed at understanding the licencing role of Plk1, independent of the work reported here, identified and evaluated the functional contribution of Plk1 phosphorylation on the subunits of the Mis18 complex (Parashara et al., bioRxiv 2024). Serendipitously, this recent work has now validated our hypothesis proposed based on the structural characterisation reported here and demonstrates that a Plk1-mediated phosphorylation cascade activates the Mis18 α/β complex via a conformational switch of the N-terminal helical region of Mis18 α which facilitates a robust HJURP-Mis18 α/β interaction (Parashara et al. bioRxiv 2024). An independent study from the Musacchio lab (Conti et al., bioRxiv 2024) also reports similar findings, mutually strengthening our independent conclusions. Overall, these studies highlight the importance of the critical structural insights into the Mis18 complex this study reports. We now explicitly discuss the validation of our original hypothesis by citing our recent work along with that of the Musacchio lab. The corresponding section of the last paragraph now reads as follows (page 17 line 10): “Previously published work identified amino acid sequence similarity between the N-terminal region of Mis18 α and R1 and R2 repeats of the HJURP that mediates Mis18 α/β interaction (Pan et al., 2019). Deletion of the Mis18 α N-terminal region enhanced HJURP interaction with the Mis18 complex (Pan et al., 2019). Here, we show that the N-terminal helical region of Mis18 α makes extensive contact with the C-terminal helices of Mis18 α and Mis18 β , which had previously been shown to mediate HJURP binding by Pan et al., 2019. Collectively these observations suggest that the N-terminal region of Mis18 α might directly interfere with HJURP - Mis18 complex interaction. Two independent recent studies (Parashara et al., 2024, Conti et al., 2024) reveal that this is indeed the case and a Plk1-mediated phosphorylation cascade involving several phosphorylation and binding events of the Mis18 complex subunits relieve the intramolecular interactions between the Mis18 α N-terminal helical region and the HJURP binding surface of the Mis18 α/β C-terminal helical bundle. This facilitates robust HJURP-Mis18 α/β interaction *in vitro* and efficient HJURP centromere recruitment and CENP-A loading in cells. Overall, these studies also highlight the importance of the critical structural insights into the Mis18 complex we report here.”

3. I am happy with the way cell biology data and the methods are presented so that they can be reproduced. The experiments are adequately replicated and the statistical analysis adequate. It will help to include sample size of cells or centromeres used for building the graphs.

We have now included this information in figure legends of Fig. 3a, 3c, 4b, 4c, 5b, 5c and 5d.

4. This is a strong interdisciplinary study using a variety of in vitro and in vivo techniques. Can

the authors discuss if they expect chromatin associated Mis18 complex to host a similar structure as the soluble one? In other words, are they able to comment on any key differences between chromatin and non-chromatin associated Mis18 complexes.

We thank the reviewer for the suggestion. We agree that one cannot rule out the possibility of the Mis18 complex undergoing compositional and/or conformational variations during the processes of CENP-A loading at centromeres. We now explicitly discuss this possibility in the last paragraph of the discussion section (page 18 line 10): “Future work aimed at characterising the intermolecular contact points between the subunits of the Mis18 complex, centromeric chromatin and CCAN components and understanding if the Mis18 complex undergoes any conformational and/or compositional variations upon centromere association and/or during CENP-A deposition process, will be crucial to delineate the mechanisms underpinning the centromere maintenance.”

Minor comments: -

In cell biology experiments, fluorescence intensities could be presented as a superplot for added value across cells and repeats (instead of bar graphs). More on superplot: <https://doi.org/10.1083/jcb.202001064>.

We thank the reviewers for this kind suggestion. We have now included graphs made using ‘superplot’ as suggested.

In general, ACA levels do not appear to change significantly between WT and mutant expressing cells although new CENP-A loading is significantly absent in the presence of a few mutants - please comment if ACA used here can recognise CENP-A. Would this mean that old CENP-A remains normally?

We thank the reviewer for this comment. While new CENP-A incorporated at centromeres is selectively labelled using the SNAP-tag, the ACA antibody used in these experiments can recognise CENP-A, CENP-B and CENP-C, with CENP-B being the primary target (Kallenberg, Clinical Rheumatology, 1990). We would also like to note that ACA has commonly been used to locate the centromere in CENP-A loading assays where new CENP-A levels are assessed via selective labelling (e.g. McKinley 2014).

It is unclear whether any of the mutant acted in a dominant negative fashion in the presence of endogenous Mis18 proteins. It would have been useful to test this particularly in the context of mis18alpha mutants that seem to fully abolish new CENP-A recruitment.

As Mis18 subunits oligomerise (homo and hetero), we thought expressing these mutants in the presence of endogenous proteins might interfere with endogenous protein in a heterogenous manner and might make the interpretation difficult. Hence, we did not test this. Instead, as described in the manuscript we have tested these mutants in siRNA rescue experiments (Fig. 3, 4 and 5).

Full Revision

In figure 3a, GFP panel (input lane, 1) is shown to mark a band corresponding to GFP. Is this expected? Please comment.

Yes, as a control, an empty vector was transfected to express just GFP along with Mis18 α -mCherry. These were used to show that there was no unspecific interaction between the beads used for IP or Mis18 α -mCherry and GFP tag, and that any interaction seen was due to Mis18 β . A similar control was used in S4b, where mCherry was expressed along with Mis18 β -GFP. We have now clarified this in the corresponding legends of Fig. 4a and S4b.

Would be useful to have the scale for the cropped images presented as insets. Figure 4B should read YFP and not YPF.

We apologise for this typographical error. We have now corrected this.

The authors may want to explain whether the tag differences matter for their study (Case in point: His-SUMO-Mis18a191-233 WT and mutant His-MBP-Mis18b188-229 proteins).

The MBP tag was chosen to perform amylose pull-down assays, whereas the SUMO tag was chosen to increase the protein size. This is crucial as the C-terminal fragments of Mis18 α and Mis18 β are less than 50 amino acids long and are not easy to visualise by the band intensity in the Coomassie-stained SDS PAGE gels.

Reviewer #3 (Significance (Required)):

This work elucidates the structural basis of Mis18 complex assembly and the intermolecular interfaces essential for Mis18 functions. This is a significant advance in the field as it helps researchers in the field better understand CENP-A deposition and mechanism underpinning the maintenance of centromere identity. This is a broad area of research benefitting those studying cell division, genome stability, centromere identity and epigenetics might all be interested in and influenced by these findings. Novelty and strength lies in combining structural and cell biology work.

Strengths of the work are structural details of the Mis18 complex. Minor weakness is the link between Mis18 structure and Centromere inheritance is limited to one immunostaining assay (I have mentioned this as a minor comment because addressing this may not be within the scope of this manuscript and is likely to require a repeat of a vast majority of the work with additional reagents which may not directly add value to the current manuscript).

Dear Prof. Arulanandam,

Thank you for submitting your revised manuscript, which was previously peer reviewed at Review Commons. An arbitrating advisor has now evaluated the revised manuscript. The advisor finds that the initial concerns of referees were adequately addressed. He/she thus recommends publication in EMBO Reports.

However, I need you to address the points below before I can accept the manuscript.

- We can accommodate up to 5 keywords. Therefore, please remove 4 of the keywords.
- Please remove the Author Contributions section from the manuscript.
- Please add a "Disclosure Statement and Competing Interests" section (<https://www.embopress.org/page/journal/14693178/authorguide#conflictsofinterest>).
- We note a discrepancy in regarding one of the author names - A. Arockia Jeyaprakash in the manuscript file vs. A. Jeyaprakash Arulanandam in the manuscript submission system.
- As per our format requirements, in the reference list, citations should be listed in alphabetical order and then chronologically, with the authors' surnames and initials inverted; where there are more than 10 authors on a paper, 10 will be listed, followed by 'et al.'. Please see <https://www.embopress.org/page/journal/14693178/authorguide#referencesformat>
- Along similar lines, please update the reference style of preprints as follows:
 - o In-text citation: (preprint: NAME1 et al, YEAR)
 - o Author NAME1, Author NAME2, (YEAR) article title. bioRxiv doi: 1234/002.dj123 [PREPRINT]
- Please fill out and include an author checklist as listed in our online guidelines (<https://www.embopress.org/page/journal/14693178/authorguide>)
- We note that the panels of Figure 2 (a-f) are not individually callout out in the text.
- Please rename Table S1 and Table S2 as Table EV1 and Table EV2 and submit them as such. Please remember to update the callouts in the text accordingly.
- The manuscript sections should be in the following order: Title page - Abstract & Keywords - Introduction - Results - Discussion - Methods - Data Availability - Acknowledgments - Disclosure Statement & Competing Interests - References - Figure Legends - Tables with legends - Expanded View Figure Legends.
- Please submit source data as per the email from our Source Data Coordinator Dr. Hannah Sonntag dated 15.04.2024.
- Supplementary Figures, legends, callouts, etc. need to be renamed to EV Figures: Figure EV1, Figure EV2, etc.
- Our production/data editors have asked you to clarify several points in the figure legends:
 - o Please note that the specific URLs for 7SFY and 7SFZ datasets are not provided in the data availability statement.
 - o Please note that the accession ID for the ProteomeXchange Consortium via the PRIDE database is not provided in the data availability statement.
 - o Please note that in figures 5c-d; there is a mismatch between the annotated p values in the figure legend and the annotated p values in the figure file that should be corrected.
 - o Please note that the scale bar needs to be defined for figure 5c.
- The manuscript sections should be in the following order: Title page - Abstract & Keywords - Introduction - Results - Discussion - Methods - Data Availability - Acknowledgments - Disclosure Statement & Competing Interests - References - Figure Legends - Expanded View Figure Legends.
- Papers published in EMBO Reports include a 'synopsis' and 'bullet points' to further enhance discoverability. Both are displayed on the html version of the paper and are freely accessible to all readers. The synopsis includes a short standfirst summarizing the study in 1 or 2 sentences (max 35 words) that summarize the paper and are provided by the authors and streamlined by the handling editor. I would therefore ask you to include your synopsis blurb and 3-5 bullet points listing the key experimental findings.
- In addition, please provide an image for the synopsis. This image should provide a rapid overview of the question addressed in the study but still needs to be kept fairly modest since the image size cannot exceed 550 (width) x 300-600 (height) pixels.

Thank you again for giving us to consider your manuscript for EMBO Reports, I look forward to your minor revision.

Kind regards,

Deniz Senyilmaz Tiebe

--

Deniz Senyilmaz Tiebe, PhD
Editor
EMBO Reports

Rev_Com_number: RC-2023-02275

New_manu_number: EMBOR-2024-59255V1-T

Corr_author: Arulanandam

Title: Structural Basis for Mis18 Complex Assembly: Implications for Centromere Maintenance

The authors have addressed all minor editorial requests.

Dear JP,

Thank you for submitting your revised manuscript. I have now looked at everything and all is fine. Therefore, I am very pleased to accept your manuscript for publication in EMBO Reports.

Congratulations on a nice work!

I need your input on one more point before we can transfer your manuscript to our production team. We discourage usage of punctuation marks in the title. Therefore, could you please propose a different title without the colon? It should not exceed 100 characters including spaces. I look forward to hearing from you. Thanks a lot.

Kind regards,

Deniz

Deniz Senyilmaz Tiebe, PhD
Editor
EMBO Reports

--

Rev_Com_number: RC-2023-02275

New_manu_number: EMBOR-2024-59255V2

Corr_author: Jeyaprakash

Title: Structural Basis for Mis18 Complex Assembly: Implications for Centromere Maintenance